

# High-Resolution Regional Climate Modeling and Projection over Western Canada using a Weather Research Forecasting Model with a Pseudo-Global Warming Approach

Yanping Li[1,2], Zhenhua Li[1], Zhe Zhang[1,2], Liang Chen[3,1], Sopan Kurkute[1,2], Lucia Scaff[1,2], Xicai Pan[4,1]

[1]Global Institute for Water Security, University of Saskatchewan, Saskatoon, SK, Canada
[2]School of Environment and Sustainability, University of Saskatchewan, Saskatoon, SK, Canada
[3]CAS Key Laboratory of Regional Climate Environment for Temperate East Asia, Institute of Atmospheric Physics, Chinese Academy of Sciences (CAS), Beijing, China
[4]Institute of Soil Science, Chinese Academy of Sciences (CAS), Nanjing, China

*Correspondence to*:  Yanping Li (Yanping.li@usask.ca); Zhenhua Li(zhenhua.li@usask.ca)

**Abstract.**  To assess the hydroclimatic risks posed by climate change in western Canada, this study conducted a retrospective simulation (CTL) and a pseudo-global warming (PGW) dynamical downscaling of future warming projection under RCP8.5 from an ensemble of CMIP5 climate model projections using a convection-permitting 4-km Weather Research Forecasting (WRF) model. The convection-permitting resolution of the model avoids the error-prone convection parameterization by explicitly resolving cumulus plumes. The evaluation of surface air temperature by the retrospective simulation WRF-CTL against a gridded
observation ANUSPLIN shows that WRF simulation of daily mean temperature agrees well with ANUSPLIN temperature in terms of the geographical distribution of cold biases east of the Canadian Rockies, especially in spring. Compared with the observed precipitation from ANUSPLIN and CaPA, the WRF-CTL simulation captures the main pattern of distribution, but with a wet bias seen in higher precipitation near the British Columbia coast in winter and over the immediate region on the lee side of the Canadian Rockies.  The PGW simulation shows more warming than CTL, especially over the polar region in the northeast, during the cold
season, and in daily minimum temperature. Precipitation changes in PGW over CTL vary with the seasons: In spring and late fall for both basins, precipitation is shown to increase, whereas in summer in the Saskatchewan River Basin, it either shows no increase or decreases, with less summer precipitation shown in PGW than in CTL for some parts of the Prairies. This seasonal difference in precipitation change suggests that in summer the Canadian Prairies and the southern Boreal Forest biomes will likely see a slight decline in precipitation minus evapotranspiration, which might impact soil moisture for farming and forest fires. With almost no
increase in summer precipitation and much more evapotranspiration in PGW than in CTL, the water availability during the growing season will be challenging for the Canadian Prairies. WRF-PGW shows an increase of high-intensity precipitation events and shifts the distribution of precipitation events toward more extremely intensive events in all seasons, as current moderate events become extreme events with more vapor loading, especially in summer. Due to this shift in precipitation intensity to the higher end in the PGW simulation, the seemingly moderate increase in the total amount of precipitation in summer for both the Mackenzie and
Saskatchewan river basins may not reflect the real change in flooding risk and water availability for agriculture. The high-resolution downscaled climate simulations provide abundant opportunities both for investigating local-scale atmospheric dynamics and for studying climate impacts in hydrology, agriculture, and ecosystems. The change in probability distribution of precipitation intensity also calls for innovative bias-correction methods to be developed for the application of the dataset when bias-correction is required.

## 1  Introduction

Climate change has been increasingly evident as shown by the rising global average surface temperature of the earth, despite some natural interannual and decadal fluctuations, since the instrumental records of temperature started in 19th century (Bindoff et al.,



2013; IPCC, 2013). Climate change and its potential risks to the environment and society have become one of the most pressing issues for humanity. As greenhouse gas (GHG) emissions continue to rise due to human activities, the global mean temperature will increase, consequently, so will climate extremes (Easterling et al., 2000; Karl et al., 2006; Sugiyama et al., 2009). The changing climatological mean and increasing extremes could impact many aspects of the ecosystem, environment, and society. Although consensus about climate change has been established, how the climate system will respond to potential GHG radiative forcing is

less clear due to the complexity of the climate system. Even more unclear is how the regional climate and hydrology will respond to a specific warming climate scenario. This challenge to project a regional climate response is due not only to the complexity of atmosphere, ocean, land surface, and hydrological processes themselves, but also to the numerous interconnections, interactions, and types of feedback between each component of the climate system.

Numerical models, supported by comprehensive observation validations, are indispensable tools to enhance our knowledge of the

climate system and to make climate projections. Since the industrial revolution, Global Climate Models (GCMs) have been widely used to assess the climatic impacts of accumulated GHG emissions and to project the future climate under different emission scenarios. For example, the Coupled Model Intercomparison Project Phase 5 (CMIP5) comprises more than 20 model centers and more than 60 GCM combinations. CMIP5 uses a standard set of model simulations to evaluate how realistic the GCMs are in simulating the recent past and also provides multiple scenario projections of future climate changes in the near term (out to about

2035) and long term (out to 2100 and beyond).

GCMs include a multitude of processes with a gamut of temporal-spatial scales. To represent the complex climate system in numerical models, processes ranging from scales as small as aerosols and turbulence to those as large as the planet, e.g., the continental drift in paleoclimate simulation, have to be formulated explicitly or through parameterization. To faithfully represent the basic energy balance of the planet, GCMs need to simulate the planetary scale climate processes that transfer heat and mass

through extensive ocean currents and jet streams. In addition to this large-scale advection in the atmosphere and oceans by mean flow, GCMs also need to simulate the atmospheric and oceanic eddies embedded in the flow that transport a massive amount of heat meridionally. These eddies, which rise from the thermal gradient, are bound to evolve as the global temperature rises and alters the tropic-polar thermal gradient.

The climate system also has multiple-year oscillations (e.g., the El-Nino Southern Oscillation) and multi-decadal oscillations (e.g.,

the Pacific Decadal Oscillation and Atlantic Meridional Oscillation), which often obscure the secular trend (Xie and Kosaka, 2017). To average out the natural oscillations in the climate system and to reach equilibrium for the slow processes (e.g., deep ocean circulation, permafrost), GCM usually needs to perform simulations for periods from decades to centuries. Due to high computation costs, the large spatial and temporal scales that GCMs have to capture compel them to settle on coarse resolutions. Thus, GCMs have to represent the effects of small-scale processes such as convection, gravity waves, and turbulent transport through

parameterization.

However, climate impacts on the ecosystem and human society often occur on local and regional scales, both of which are important for climatic impacts. For example, surface air temperature is strongly affected by underlying surface and local circulation. To bridge the gap between large-scale projection and local-scale climatic impact, regional climate downscaling is often performed on GCM projections. Statistical downscaling has the advantage of being computationally cheap and easy to implement but suffers

from the assumption of the stationarity of the statistical distribution of the hydrometeorology variables. In an ever-changing climate and earth system, stationarity is not a norm but an exception. Dynamical downscaling using regional climate models (RCM) can provide added value to the understanding of regional climate change by explicitly representing some of the small-scale processes that are critical but poorly represented in GCMs (Castro, 2005).



The added values of RCM simulations relative to driving GCMs are widely accepted, especially in regions with a strong heterogeneous underlying boundary and for mesoscale atmospheric processes, in particular, when the RCM is constrained at the large spatial scales through boundary conditions and spectral nudging (Feser et al., 2011). RCM simulations are especially valuable for variables such as near-surface temperature and humidity, which are strongly affected by the representation of near surface processes. The mesoscale phenomena such as polar lows (Feser et al., 2011) and mesoscale convective systems (Prein et al., 2017a)

can be represented more realistically in RCM simulations. Because RCMs can resolve subgrid-scale processes in GCMs, which are important to water cycles and the ecosystem, they are widely used to provide detailed projections of future climate scenarios and downscaling information for impact studies, especially those associated with the aforementioned fine-scale processes.

RCMs have been individually applied to downscale temperature and precipitation projection over North America and under inter-comparison frameworks such as NARCCAP (Mearns et al., 2009, 2015) and CORDEX (Giorgi et al., 2009). These inter-

comparison frameworks provide a glimpse into the uncertainties in regional climate downscaling through a common combination of driving GCMs, RCMs, and multiple emission scenarios. The horizontal resolutions of RCMs used in the recent coordinated regional climate downscaling efforts are usually larger than 10 km. With these relatively coarse resolutions, RCMs still have to rely on convection parameterization to represent deep convection in the models.

In climate simulation, convection parameterization is a major source of errors, which is used to represent the statistical effects of

subgrid cumulus plumes on the redistribution of mass, heat, and momentum on the grid-scale mean flow. Convection parameterization used in GCMs and coarse-resolution RCMs causes bias in the simulated hydrological cycle: underestimated dry days, misrepresentation of the diurnal cycles of convective precipitation, etc. Deep convection, however, contributes to a relatively large percentage of precipitation amounts and extremes, especially during warm seasons. Poor simulation of deep convection is a stubborn problem for RCMs in climate projection and regional climate dynamical downscaling. One way to avoid the errors

introduced by convective parameterization is to resolve the cumulus plumes explicitly with high resolution models. RCMs with horizontal grid spacing less than 4 km can resolve convective processes and are often referred to as convection-permitting models (CPMs). As well as explicitly representing deep convection, CPMs also permit a more accurate representation of underlying surface and topography. As computing capability grows, CPMs, or cloud-resolving models emerge as a promising tool to generate more realistic regional to local scale climate simulations compared to models with coarser resolution and convective parameterization

(Prein et al., 2015). Although CPMs require higher computational resources than lower resolution models, the computing costs of CPMs can be justified by their ability to simulate mesoscale convective systems more realistically and to produce better convective and orographic precipitation (Prein et al., 2015; Weusthoff et al., 2010).

CPMs have great benefits for dynamical downscaling over western Canada due to its geographic characteristics. Most notably, western Canada features the Canadian Rockies, where steep terrain and small-scale atmospheric processes play important roles in

wave dynamics and mountain meteorology. In cold seasons, especially, the atmosphere, hydrology, and cryosphere strongly couple with each other through small-scale boundary layer processes, including snow cover, snow melt, and blowing snow. On the other hand, western Canada also encompasses the Canadian Prairies, where climate downscaling seems straightforward because of its seemingly homogeneous landscape. However, in the Prairies summer convections contribute the most precipitation, and these subgrid scale convections in GCMs need to be properly simulated by using high-resolution convection permitting models.

To provide high-resolution convection-permitting downscaling for western Canada, a set of 4-km convection-permitting WRF simulations was conducted for the current climate and the high-end emission scenario of RCP8.5. The 4-km convection-permitting retrospective simulation (CTL, October 2000-September 2015) was driven by ERA-interim reanalysis (Dee et al., 2011). The future climate sensitivity simulation was conducted using reanalysis-derived initial and boundary conditions for the same period as CTL but perturbed with changes in field variables derived from the CMIP5 ensemble-mean high-end emission scenario (RCP8.5)



climate projections, the so called Pseudo-Global Warming(PGW) method. In this paper, we evaluate the performance of the retrospective simulation and investigate the dynamically downscaled regional climate change over western Canada, especially the Mackenzie River Basin (MRB) and Saskatchewan River Basin (SRB). The paper is organized as follows: section 2 introduces the model setup and data; section 3 evaluates the retrospective simulation (CTL) against observation; section 4 describes the projected climate change by the PGW versus CTL; section 5 shows the changes in temperature and precipitation extremes; section 6 discusses

the results, and section 7 summarizes the results and concludes the paper.

## 2    Model Setup and Data

### 2.1    Model Setup

The Weather Research and Forecasting (WRF) model Version 3.6.1 was used to simulate the historical (2000-2015) and projected climate (RCP8.5) over western Canada with a convection-permitting resolution of 4 km. The WRF model is fully compressible

and nonhydrostatic and uses the Advanced Research WRF (ARW) dynamical solvers. The model domain is composed of 699 x 639 grid points with 4-km horizontal resolution to cover western Canada from British Columbia and the Yukon to the west and the Mackenzie River Basin (MRB), and the Saskatchewan River Basin (SRB) to the east as shown in Fig. 1. In total, the model domain covers 2800 km in the east-west direction and 2560 km in the north-south direction. The model's vertical coordinate comprised 37 stretched vertical levels topped at 50 hPa in the lower stratosphere. The model simulations employed several

parameterization schemes, including Thompson microphysics scheme (Thompson et al., 2008), the Yonsei University (YSU) planetary boundary layer scheme, the Noah land surface model (Chen and Dudhia, 2001), and the CAM3 radiative transfer scheme (Collins et al., 2004). The deep cumulus parameterization was turned off because with a 4-km horizontal resolution the model can explicitly resolve deep convection and simulate convective storms. The convection-permitting model produces precipitation more realistically by directly resolving convections. As well, because using cumulus parameterization schemes at this resolution often

produces unrealistic convection (Westra et al., 2014), cumulus parameterization was switched off.  Subgrid cloud cover was also disabled.

### 2.2    Numerical experiments

Two 15-year WRF simulations were conducted to simulate the regional climate under the historical and future climate using reanalysis and climate change forcing derived from CMIP5 ensembles, respectively. The control experiment (CTL), a

retrospective/control simulation, aimed to reproduce the current climate statistics in terms of variability and mean state from October 1, 2000 to 30 September 2015. This control simulation was forced using 6-hourly 0.7 degree ERA-Interim reanalysis data (Dee et al., 2011) directly. WRF simulation was directly forced by 4-km one-way nesting without an intermediate buffering coarse grid between the ERA-Interim reanalysis and WRF domain because the ~75 km resolution reanalysis was shown to be adequate (Liu et al., 2017). The second simulation was a climate perturbation or sensitivity experiment following the Pseudo-Global

Warming (PGW) approach used in Colorado-Headwaters work (Rasmussen et al., 2014, 2011). Climate projections from GCMs introduce large uncertainties because of the substantial inter-model variability among GCMs (Deser et al., 2012; Mearns et al., 2013), which can obscure the climate change response due to global warming. Using the PGW approach, rather than the inter-model variability, can isolate radiative forcing and its associated circulation as the sole reason for the regional climate response.

Regional climate downscaling using convection-permitting models has a range of advantages over using models that rely on

convection parameterization, including better convective precipitation simulation and the ability to compare regional climate





changes directly related to global warming scenarios. Due to these benefits, convection-permitting PGW simulation (Liu et al.,
2017) has been used in several recent studies to investigate the intensification of hourly precipitation extremes (Prein et al., 2017b),
the decrease of overall precipitation frequency and light-moderate precipitation events over the contiguous United States (CONUS)
(Dai et al., 2017), and the increase of rain-on-snow events in western North America (Musselman et al., 2018). The PGW forcing
was derived from climate change signals from a 19-member ensemble mean of CMIP5 models. In particular, PGW 15-year (2000-

2015) simulation was forced with the same period of 6-h ERA-Interim reanalysis as in CTL, plus a climate perturbation from the
ensemble CMIP5 RCP8.5 projection:

PGW_forcing = ERA-Interim + $\Delta_{CMIP5rcp8.5}$ ,                    (1)

where $\Delta_{CMIP5rcp8.5}$ is the climate change signals derived from the CMIP5 multi-model (19 ensemble members) ensemble-mean
under the RCP8.5 emission scenario from 2071-2100 relative to 1976-2005. The choice of the model members and the details of

the ensemble members of the 19 CMIP5 models are provided in Liu et al. (2017).

Climate change signals are interpolated according to calendar date using the monthly $\Delta_{CMIP5rcp8.5}$ data for both surface variables
and three-dimensional field variables. The surface variables such as surface temperature, soil temperature, sea level pressure, and
sea ice are incorporated into the PGW forcing by including the climate changes signals in the initial and boundary conditions for
CTL. Similarly, PGW forcing perturbations were also added to the three-dimensional field variables, such as horizontal wind

components, air temperature, specific humidity, and geopotential in the initial and boundary conditions of CTL.

The climate change signals in Fig. 2 show the circulation and thermodynamic changes in the PGW forcing for different seasons.
As shown in the third row of Fig. 2, the temperature increases at 750 hPa in the lower troposphere under RCP8.5. The warming is
larger in the northwest and in the MRB than in the southwest and in the SRB, especially in autumn and winter. The warming ranges
from 3 to 4℃ in winter and spring and from 4 to 5℃ in summer and autumn. Accompanying this warming is a moderate decrease

(0.5 to 2 percent) of relative humidity throughout the domain, with a larger decrease in the south in summer and autumn. In winter
the pattern of relative humidity changes is more complex, with larger decreases in the east. In spring, the relative humidity changes
follow a pattern that generally aligns with the Canadian Rockies; the Canadian Prairies and the MRB experience larger decreases
of relative humidity than the southwest of the domain.

As shown in the third row of Fig. 2, the change of geopotential height (GPH) at 750 hPa presents a pattern as thickness between

the lower atmospheric isobaric surfaces, consistent with the temperature change, as the thickness is proportional to the average
temperature of the layer. The warming in the lower atmosphere causes a general increase of GPH throughout the domain, whereas
the stronger warming in the northeast corresponds to the larger increase of GPH in the northeast. The largest increase of
geopotential height occurs in autumn and winter and shows a magnitude of 40 m. The smallest increase of 35 m occurs in winter
in the southwest along the British Columbia coast. Accompanying this pattern of change in GPH, there is a weakening of the

westerly flow in all seasons in the order of .5 to 1 ms$^{-1}$ at 750 hPa due to geostrophic balance.

At the mid-troposphere level, the general pattern of change in GPH at 500 hPa is similar to that at 750 hPa but with larger values
of 90-100m, as shown in the second row of Fig. 2. Consistent with this geopotential height pattern, the change in the wind field is
similar to that at 750hPa but with larger magnitudes, especially in winter and spring. Temperature increases are also slightly larger
than those at 750 hPa, with about 5-6℃ at 500 hPa in summer and autumn. In contrast, in winter and spring, the warming at 500

hPa is less than that of 750 hPa.

For the upper level at 250hPa, the increase in temperature ranges from 1 to 4℃, with stronger warming in the south, as shown in
the top row of Fig. 2. The warming at 250hPa is less than that at the lower levels, especially for the cold seasons, when the warming
is only about 1℃. The geopotential height experiences the largest increase in summer and the smallest increase in winter. In both



summer and winter, larger increases of geopotential height in the central and eastern domain correspond to the west-east geopotential height gradient and to a southerly wind increase in the northern part of the domain. In winter (summer), there is an increase (decrease) of westerly wind in the southern part. In spring there is an increase of westerly wind in the southern domain and an increase of southerly in the west. In autumn, there is an increase of southwesterly wind in the northwest, while in the Canadian Prairies an increase of easterly wind occurs.

**2.3    Verification Data**

The simulation evaluation was conducted against two gridded datasets for temperature and precipitation for the retrospective CTL simulation from 2000 to 2015. The NCEP North American Regional Reanalysis (NARR) (Mesinger et al., 2006) and the surface station observations from Environment Climate Change Canada were also used in basin average evaluations.

**2.3.1    ANUSPLIN**

ANUSPLIN was first used to develop a high spatial resolution (~10 km) data set of daily precipitation and minimum and maximum temperature for the period 1961–2003 for Canada (Hutchinson et al., 2009). ANUSPLIN uses a thin-plate smoothing spline algorithm composed of the spatially continuous functions of latitude, longitude, and elevation (Hutchinson et al., 2009). The algorithm offers an efficient way to develop spatially continuous climate distribution for temperature and precipitation (Xu and Hutchinson, 2013). Hopkinson et al. (2011) further improved the Canadian ANUSPLIN data through reducing significant residuals
by aligning the climatological day at observation stations and expanding the gridded dataset to cover 1950–2011. The Canadian ANUSPLIN has been updated to 2017 and used to evaluate gridded climate models and reanalysis datasets (Eum et al., 2012) and to compare the impacts of different climate products on hydro-climatological applications (Bonsal et al., 2013; Eum et al., 2014; Wong et al., 2016). Our evaluation of CTL performance uses daily temperature, maximum temperature, minimum temperature, and precipitation from ANUSPLIN Canada.

**2.3.2    CaPA**

The Canadian Precipitation Analysis (CaPA) data set is a precipitation reanalysis with high spatial resolution (~15 km) and 6-hourly temporal resolution. CaPA is derived from various sources of precipitation data such as station observation, satellite remote sensing, weather radar, and short-term forecasts from the Global Environmental Multiscale (GEM) model (Mahfouf et al., 2007). The short-term precipitation forecasts from the Canadian Meteorological Centre (CMC) regional GEM model were used as the
background field with the rain-gauge measurements from the National Climate Data Archive as the observations to generate an analysis error at every grid point (Mahfouf et al., 2007). CaPA's optimum interpolation method depends on three key parameters to specify the error statistics: background error, observation error, and characteristic length scale. The error statistics from observations and the background field were then used in the optimum interpolation technique to generate 6-hourly precipitation data. A recent paper by Fortin et al. (2018) presents a summary on the development and applications of CaPA in the last decade.

**2.3.3    NARR**

NCEP North American Regional Reanalysis (NARR) uses the NCEP Eta Model together with the Regional Data Assimilation System to assimilate precipitation along with other variables. In NARR precipitation observations are assimilated using latent heating profiles (Mesinger et al., 2006). NARR data are available from October 1978 to November 2018 at a relatively high spatial





and temporal resolution: 32-km grid spacing, 45 vertical layers, and 3-hour time intervals. The NARR dataset is used only for
       comparing basin average temperature and precipitation for SRB and MRB.

## 3      Evaluation of CTL Experiment

### 3.1      Near Surface Temperature

       Surface air temperature is a key meteorological variable that directly affects the daily life of human beings, physiological
development of field crops, agricultural product quality, and various hydrological processes. For humans, extreme and persistent
       hot days in summer can cause health issues including heat cramps, heat exhaustion, and heat stroke, especially for vulnerable
       populations such as the elderly. For agriculture, extreme hot spells of multiple days with a maximum temperature hovering above
       the cardinal maximum, the temperature at which crop growth ceases, can significantly reduce crop yields. At the other extreme,
       the effects of very cold temperatures range from a minor inconvenience for some to  severe infrastructure damage and increased
mortality for vulnerable populations. As the mean temperature changes, the extreme distribution of temperature also changes
       substantially, sometimes more than the changes in the mean. From the perspective of hydrology, surface air temperature's
       simulation is also crucial to obtain realistic evapotranspiration, energy exchange between the surface and atmosphere, and phase
       transition of water near the ground. Because of all these temperature effects, evaluating the surface air temperature simulation is
       critical in laying the foundation for applying the WRF-CTL and PGW simulations to hydrological modeling, climate projection,
and climate change impact analysis.

### 3.1.1      Mean Temperature

       The comparison of surface air temperature (2m) between CTL and ANUSPLIN in Fig. 3 shows that WRF simulation of daily mean
       temperature agrees well with ANUSPLIN temperature in terms of the geographical distribution of cold biases east of the Canadian
       Rockies, especially in spring. The spring (March, April, May), summer (June, July, August), autumn (September, October,
November) and winter (December, January, February) from WRF-CTL and the gridded observation analysis ANUSPLIN are
       presented in Fig. 3. Both ANUSPLIN and CTL show a consistent spatial distribution and seasonal change of temperature gradient.
       In spring there is a strong cold bias (about -5℃) over the Canadian Prairies, with a small warm bias of 1-2℃ in the northeast
       domain. In summer the hottest region is located in the southern Canadian Prairies, with temperatures decreasing toward northeast
       and coastal regions. In autumn the temperature in both ANUSPLIN and WRF decreases from the southern border to the Arctic.
However, there are a few noticeable biases in the simulated daily mean temperature. In winter and spring, the temperature decreases
       from southwest British Columbia toward northeast of the domain as the regional climate changes from oceanic to subarctic. There
       is a significant warm bias (about 5-6℃) in winter near the Yukon and western Northwest Territories. In winter small warm biases
       (about 2℃) occur in central and northern British Columbia. There are also small cold biases (-1 - -2℃) in all seasons east of the
       Canadian Rockies. Due to these biases in winter and spring, the WRF-CTL simulation tend to enhance the temperature difference
between the warmer regions near the Pacific coast and the colder Canadian Prairies. Although regional climate models are forced
       by reanalysis data on the boundary and underlying surface, the near surface temperature is strongly affected by the representation
       of surface processes and boundary layer energy exchange. The cold bias in spring over the Canadian Prairies is caused by several
       factors: a wet bias precipitation and cold bias in temperature in winter, and the overestimation of snow cover in the region, which
       amplifies the cold bias in spring through snow-albedo feedback.





### 3.1.2 Daily Minimum and Maximum Temperature

The daily minimum temperature ($T_{min}$) of WRF-CTL and ANUSPLIN (Fig. 4) shows a similar geographical distribution to that of the daily mean temperature in all seasons. The main difference between the $T_{min}$ distribution and daily mean temperature distribution is that the south-north temperature gradient becomes less in summer. Compared to the bias of daily mean temperature, WRF-CTL simulation of $T_{min}$ relative to ANUSPLIN shows a stronger warm bias in the northwest (the Yukon and western Northwest Territories), with a magnitude of 4℃ in winter. Additionally, the cold bias of CTL in $T_{min}$ over the Prairies in spring decreases by 50% compared to that of the daily mean temperature (about -2-4℃ vs -6℃).

The daily maximum temperature for four seasons by WRF and ANUSPLIN is shown in Fig. 5. The cold bias in the Prairies during spring shown in the $T_{max}$ is more pronounced (> 6℃) in the daily mean temperature. The warm bias in the northeast in spring is also stronger. The $T_{max}$ and $T_{min}$ daily minimum bias distribution shows that the cold bias in spring in the Prairies is stronger in the early afternoon, when there is strong solar insolation, and much weaker at night. This cold bias in spring may relate to a combination of the overestimation of snow cover and the albedo biases associated with improper representation of snow in the land surface model (Meng et al., 2018).

### 3.2 Precipitation

Water resources are of strategic significance for the environment, agriculture, and society, especially for semi-arid regions in most of western Canada. Precipitation is an important component of water balance and is essential for hydrological modeling as all runoff comes from precipitation, either directly or indirectly. The ability of climate models to capture the temporal-spatial characteristics of observed precipitation is crucial for their application as input for hydrological models. Although ANUSPLIN is a gridded observation dataset, its coverage over the northern part of western Canada relies on a very limited number of stations. CaPA, a reanalysis dataset, has been shown to have better overall spatial distribution of precipitation than ANUSPLIN (Fortin et al., 2018). Therefore, we compared WRF-CTL's precipitation to that of both ANUSPLIN and CaPA.

As shown in Fig. 6 and 7, the WRF-CTL simulation captures the main precipitation distribution pattern in the observed precipitation from ANUSPLIN and CaPA: high precipitation near the BC coast in winter and over the immediate region on the lee side of the Canadian Rockies. WRF-CTL's spatial pattern more closely resembles CaPA's than it does ANUSPLIN's. Both CaPA and WRF-CTL are significantly wetter than ANUSPLIN, especially in the mountainous region and northern part. Compared to ANUSPLIN, WRF-CTL's wet bias mainly resides over the mountain ranges by the Pacific Ocean and in the Canadian Rockies. This wet bias associated with topography is as high as 1.7 mm/day and more prominent in winter and spring. It must be considered, though, that gridded observation analyses often underestimate precipitation over mountains, where data is scarce, through interpolation from available lower elevation observations. East of the Canadian Rockies, there are moderate wet biases (about 0.5-0.9 mm/day) across the Prairies and the boreal forest. In terms of WRF-CTL's relative bias in reference to ANUSPLIN, there is significant wet bias (+90%) in the northern domain including the MRB for all seasons. For the SRB, a large dry relative bias occurs in winter due to low observed precipitation during this season. However, according to the evaluation by Wong et al. (2016), ANUSPLIN underestimates annual precipitation by 10% to 50% from the south to north of western Canada relative to gauge observation in the region from 2002-2012. Thus, the large wet bias of WRF-CTL relative to ANUSPLIN in the north is largely due to the large dry biases of ANUSPLIN there.

Relative to CaPA, the wet bias of WRF-CTL is generally less in magnitude and less correlated with topography because CaPA assimilates GEM forecast and remote sensing data to better represent orographic precipitation than analysis data, which rely heavily on rain gauges located at lower elevations. The wet bias along the British Columbia coastal mountain ranges and the Canadian





Rockies are prominent in spring, autumn, and winter. East of the Canadian Rockies, the wet bias is located mainly over SRB and southern MRB in spring and summer. There are also regions of dry biases in the region surrounding the MRB and SRB in spring, summer, and autumn. In winter the difference between CTL and CaPA is small east of the Canadian Rockies. It is noteworthy that, according to Wong et al. (2016), the WRF-CTL wet bias relative to CaPA's east of the Canadian Rockies may be partly attributed to CaPA's relatively small dry bias (10%).

In summary, the WRF-CTL simulation captures well the spatial distribution of precipitation in all seasons. WRF-CTL's agreement with CaPA is more widespread and consistent. There are wet biases in WRF-CTL over the mountainous region compared to both ANUSPLIN and CaPA. According to the evaluation of Wong et al. (2016), both ANUSPLIN and CaPA show wet bias in the mountainous region compared to station observation, but this may be because the stations are usually situated at low altitudes and thus fail to capture the representative areal precipitation due to the topography. East of the Canadian Rockies, WRF-CTL shows a

wet bias relative to ANUSPLIN and CaPA, although both the observation and reanalysis datasets show dry bias from 2002-2012 in the region, especially in the northern part.

### 3.3    Basin Averaged Statistics

The evaluation of the simulation over the two major river basins focuses on the model performance in simulating the seasonal and interannual variations of the two key variables for hydrology: temperature and precipitation. To validate the WRF simulation

results in SRB, we compared them with several existing observation and reanalysis products. Fig. 8 shows the time series of basin-averaged precipitation (top) and temperature (bottom) in the MRB (left) and SRB (right) for the simulation period, together with different observation and reanalysis datasets (NARR, ANUSPLIN, CaPA). The pink shading is used for temperature and aqua shading for precipitation. Fig. 9 shows the annual temperature cycle from WRF-CTL (black), NARR (red), and ANUSPLIN (blue) for the MRB. Fig. 10 shows the annual cycle of precipitation for the two basins.

### 3.3.1    Mackenzie River Basin

The WRF simulation faithfully reproduces the seasonal and interannual variations of temperature. Compared to the observation, the WRF temperature simulation is within the observation spread but on the lower end of the distribution in MRB. The WRF simulation shows a cold bias for the whole year, especially from March to July. The simulated basin averaged precipitation matches well with the observation in terms of interannual variability and seasonal cycle. This good match indicates confidence in the ability

of WRF-CTL to capture the main characteristics of precipitation regime changes year-on-year, despite biases in the total amount. The simulated precipitation shows a wet bias as the WRF-CTL curve is almost always on the top of the observation envelope. As shown on the left in Fig. 10, the mean annual cycle of precipitation over the MRB is compared between WRF-CTL, the reanalysis CaPA, NARR, and observation analysis, ANUSPLIN. Both WRF-CTL simulated and observed a precipitation peak in July. The simulated precipitation by WRF-CTL is higher than ANUSPLIN in all months and very close to NARR and CaPA, except in

summer and autumn when it is about 5mm/month wetter than NARR and CaPA on average.

### 3.3.2    Saskatchewan River Basin

The WRF simulation captures the seasonal and interannual variation of precipitation in the SRB. Compared to the observation, the WRF simulation is within the observation envelope but also on the lower end of the distribution. According to Fig. 9, the annual cycles of temperature from WRF, NARR, and ANUSPLIN show good agreement for the SRB. The WRF simulation shows a cold

bias for the whole year, especially from March to July. The cold bias for the SRB is larger than that of MRB, which is consistent



with the spatial distribution of temperature bias in Fig.5, where cold bias in the Prairies are stronger in spring over the Saskatchewan
River Basin.

In Fig. 8, the simulated monthly precipitation by WRF-CTL over SRB (solid black line) from 2001 to 2013 falls within the aqua
shading, indicating the observation spread from NARR, ANUSPLIN, and CaPA. Both the simulated and observed precipitation
peak in June, in the amount of about 60 to 90 mm, and both also show the least amount of monthly precipitation in winter, with

about 20-30 mm. The simulated basin averaged precipitation shows a similar seasonal cycle and interannual variability and as
observation, as shown in Fig. 8. Again, the precipitation simulated by WRF-CTL is closer to NARR and CaPA than it is to
ANUSPLIN over the SRB. The simulated precipitation has a wet bias for all seasons, with the WRF-CTL simulated curve almost
always at the top of the observation envelope, as shown in 10.

## 4    Pseudo-Global Warming Simulation

### 4.1    Changes Relative to CMIP5

Regional climate modeling as a dynamical downscaling tool generates not only climate projections with a higher spatial resolution
but also hydroclimatic regimes different from GCMs and statistical downscaling. These improvements can be attributed to
enhanced  representation of fine-scale processes in the atmosphere and boundary conditions. Here we compared the regional
averaged temperature and precipitation for two major river basins and for output from CMIP5 versus 4-km WRF. Fig. 11 and 12

show the temperature and precipitation changes between the 1970-2000 and 2070-2100 as projected by CMIP5 ensembles and
those of WRF-CTL (2000-2015) and PGW, with 2070-2100 climate forcing simulation.

In Fig. 11 and 12, the historical runs are shown in the red/orange columns for temperature for CMIP5/WRF-CTL; the future runs
equivalent to the end of the 21st century are shown in the light red/orange columns for the temperature of CMIP5-RCP8.5/WRF-
PGW, and the difference between the future simulation and the historical simulation are represented by the white columns. In

general, temperature will increase for all months for both CMIP5 and WRF in both basins. The temperature increases in most
months for SRB are smaller in the WRF simulation than they are in CMIP5, especially in summer. The temperature increases in
MRB are about 3 degrees smaller in WRF in December and February, and about 2 degrees smaller in summer. For MRB, the
temperature increase simulated by WRF is smaller than the CMIP5 ensemble mean for most months.

In Fig. 11 and 12, the historical precipitations are shown in the dark blue/green columns; the future precipitations equivalent to the

end of the 21st century are shown in the light blue/green columns, and the difference between the future run and the historical run
are represented by the white columns. The projected changes from the CMIP5 ensemble and WRF show seasonally dependent
differences. The precipitation increase in WRF PGW simulation is lower in May, June, September, and October (the transitional
months) and higher in December, January, July, and August compared to that in CMIP5. In the ensemble of CMIP5 RCP8.5
projection, the spring and autumn precipitation will experience a large increase, and the summer precipitation will decrease in July

and stay unchanged for June and August. In contrast to this, WRF shows a large increase of precipitation in June and smaller
decreases of precipitation in July and August, with moderate increases in other months. Due to this difference in the annual cycle
of precipitation change in SRB, the dynamical downscaling by WRF-PGW shows a similar annual cycle of precipitation in the
future with a pronounced peak in June, whereas the CMIP5 ensemble projection shifts the highest precipitation month from June
to May.

Precipitation changes projected by WRF-PGW and CMIP5 vary with the seasons, but larger increases occur in spring and late fall
for both basins. Conversely, precipitation in summer and late fall for SRB either remains unchanged or shows a decrease. This



seasonal difference in precipitation change indicates that the Canadian Prairies and the southern boreal forest biomes will likely
see a slight decline in precipitation minus evapotranspiration during the summer months, possibly affecting soil moisture for
farming and forest fires. Because the precipitation increases substantially in spring in both SRB and MRB, when combined with
large temperature increases in spring, western Canada may experience more frequent rain-on-snow events that can cause severe
flooding. This projection calls for thorough investigations that combine the high-resolution regional climate simulation and state-
of-the-art hydrological modeling to quantify the probability of catastrophic flooding in spring over western Canada (Li et al., 2017).

## 4.2    Near Surface Temperature

The daily mean temperature simulated by WRF-CTL and WRF-PGW are presented in Fig. 13. The temperature increase is larger
in the northeast domain and smaller in the southwest, generally reducing the northeast-southwest temperature gradient in CTL
climatology in all seasons. The warming is the greatest in winter with around a 10C increase in the northeast quadrant. In the
Prairie, the largest warming shift occurs in the spring compared with the other seasons. This larger warming over the Prairies is
related to the shift of the daily mean temperature increase from below freezing in early and mid-spring to above freezing,  likely
causing amplified warming through snow-albedo feedback. The mean temperature in the Yukon and NWT will be similar to those
currently experienced in Saskatchewan and Alberta in spring and summer, which has great implications for the length of the
growing season in the northern territories. The winter temperature in the coldest region of the domain will be as warm as the central
Canadian Prairies in the current climate. The higher temperatures in the boreal forest region will greatly increase the probability
of wild fire, water stress, and insect pests, threatening the boreal forest ecosystem, which could eventually be replaced by grassland
and parkland (Stralberg et al., 2018).

## 4.3    Precipitation

The comparison of WRF-PGW and WRF-CTL precipitation is shown in Fig. 14. Generally speaking, the precipitation will increase
in most of the domain. In most places, WRF-PGW show an increase of precipitation of about 15–30% in all seasons compared
with WRF-CTL.  Near the British Columbia coast the magnitude of the increase can be as large as 2 mm/day. This substantial
increase of precipitation in British Columbia's coastal mountains is related to the larger water vapor loading in PGW and the
stronger effective orographic lifting to produce precipitation in that region. The change in precipitation is the least in summer,
when parts of the Prairies receive less precipitation in PGW than in CTL. With almost no increase in summer precipitation and the
much larger evapotranspiration in the Canadian Prairies in PGW than in CTL, the water availability during the growing season
will be challenging for the Canadian Prairies. In the northeast portion of the domain, northern Manitoba and NWT, the precipitation
increase could be as large as 0.5-1 mm/day, with an increase of about 40% in autumn and winter. The Yukon and central-northern
British Columbia are expected to have a 40% increase of precipitation in winter due to the higher loading of water vapor in a
warmer climate.

## 4.4    Daily Precipitation Frequency Distribution

Both for hydrological applications and societal impacts, the temporal precipitation distribution and precipitation intensity
distribution are as important as the total amount of precipitation. For example, more high-intensity precipitation events tend to
cause flash flooding and sharp spikes of runoff, while lower effective precipitation during warm seasons increases the possibility
of drought and fire.





The probability density function of precipitation shows the distribution of precipitation amounts among both light and intense precipitation events. Fig. 15 shows the probability density function of daily precipitation for the simulation of WRL-CTL and WRF-PGW and observation from CaPA and ANUSPLIN in MRB.  In the top panel, the precipitation intensity is shown with a linear scale on the x-axis and a logarithmic scale on the y-axis for probability density function (PDF) to show the detail in high-end precipitation. Compared to that of ANUSPLIN and CaPA, the WRF-CTL simulated precipitation shows a heavy tail on the

high end of the distribution, indicating that the bias in WRF-CTL mean precipitation relative to ANUSPLIN and CaPA in MRB is largely caused by a larger number of heavy precipitation events. WRF-PGW future simulation shows an even higher distribution for extreme precipitation events, indicating that these events will become even more severe under future climate conditions. In the lower panel, both log-X and log-Y are used for precipitation and probability density, enabling us to zoom in on the probability distribution of the light-to-moderate events. The red curve, the WRF-PGW future climate simulation, is now underneath CTL,

ANUSPLIN, and CaPA curves in events lower than 5 mm/day, especially in summer (JJA). This means that MRB is expected to experience fewer moderate precipitation events in addition to an increase in the probability of high intensity precipitation.

For SRB, Fig. 16, the difference between WRF-CTL (blue curve), ANUSPLIN, and CaPA is less than that in MRB. This difference is consistent with the spatial distribution of precipitation bias in Fig. 6 and 7, where the bias in SRB is much smaller than it is in MRB. WRF-PGW shows that heavy precipitation events increase and that their distribution trends towards more extreme intensive

events in all seasons, especially in summer. Similar to MRB, there is also a decrease in moderate precipitation events, shown in the log-log plot in the second row. Due to this shift in precipitation intensity to the higher end in the PGW simulation, the seemingly moderate increase in total amounts in summer for both basins may not reflect the real change in flooding risk and water availability for agriculture. Although the total amount of precipitation is expected to increase in the future, there will be less water for agriculture because extreme precipitation will contribute more to runoff than soil moisture, reducing its accessibility to crops.

As seen in Fig 16, the intervals between light to moderate precipitation events increase, because the total summer precipitation slightly increases and heavy precipitation events significantly increase, while the atmosphere needs more time to replenish water vapor (Dai et al., 2017; Trenberth et al., 2003). Dry spells also increase in frequency because both evaporation and the intervals between precipitation events increase. The intensification of droughts will have a wide-reaching impact beyond the agricultural sector: conditions are likely to be ideal for wild-fires, like those experienced across the western provinces and territories from 2014

to 2018.

### 4.5    Hourly Precipitation Extremes

The future distribution of subdaily precipitation extremes in western Canada is of particular concern, as they can cause flash floods and landslides, which damage human infrastructure and result in injuries and deaths. Here we compare the 3-hourly precipitation rate distribution among station observation, WRF-CTL, and WRF-PGW in the two basins and then investigate the changes in the

hourly precipitation rate distribution. The 3-hourly precipitation rate is first compared to observation to evaluate the extreme precipitation simulation at 3-hour intervals. The 3-hourly precipitation histograms for extreme precipitation events in MRB and SRB are shown in Fig. 17 and 18. WRF simulations are compared with EC station observations in Fig. 1 because these station observations are closer to the ground truth than the gridded observational products for which the spatial resolution is 10 km at most and are shown to have biases (Wong et al., 2016). Only the moderate to extreme values of precipitation distribution are shown here

by cutting it off at a precipitation rate of 5 mm/3hr. WRF-CTL's precipitation distribution (the blue columns) is close to that of the station observation (the gray columns) in SRB in spring, summer, and autumn, whereas WRF-CTL produces more light to moderate precipitation events than observation in MRB in most seasons except spring. These results indicate that high-resolution These results indicate that the WRF simulation captures the local precipitation extremes in all seasons well,  except winter in SRB. WRF



also shows a wet bias in light to moderate rain events in MRB in all seasons but spring, while WRF-PGW simulations (the red
columns) show a significant increase in the frequency of high intensity rainfall events across seasons.

For MRB, the WRF-CTL 3-hourly precipitation events are much more frequent than those captured by observation in the 5 mm/3hr
range, but comparable and less frequent at a higher rate in autumn and winter. In spring WRF-CTL shows fewer extreme
precipitation events than observation. In summer WRF-CTL shows more extreme events than observation at most precipitation
bins. For autumn, spring, and winter WRF-PGW sees a significant increase, 50%, 150%, 300% for 5 mm/3hr, respectively, in
precipitation events. The change in number of extreme precipitation events in MRB in the 5-10 mm/3hr range is negligible in
summer but significant at higher precipitation rates.

For SRB, the WRF-CTL agrees well with observation in spring, summer, and autumn in terms of moderate to extreme 3-hour
precipitation events, but significantly underestimates the extreme precipitation events in winter. In spring, autumn, and winter
WRF-PGW shows significant increases in extreme precipitation events. In summer WRF-PGW shows a small decrease of
precipitation events at 5 mm/3hr and only moderate increases for higher rates. It is also worth mentioning that extreme events are
much more numerous in SRB than in MRB, especially in spring and summer because the seasonal mean precipitation is higher in
SRB.

Fig 19 shows the changes of hourly precipitation distribution between surface observation and our simulations at 1-hour interval.
The black line represents observation data collected from 232 surface stations in SRB and MRB from Environment Climate Change
Canada (Website: http://climate.weather.gc.ca/index_e.html). The blue and red bars are the closest grid points to these stations
extracted every hour (in total 113952 time steps in 13 years) from the WRF domain for CTL and PGW runs, respectively. Despite
the spatial scarcity and data quality associated with station observation, the results do provide some evaluation of the WRF
simulated hourly rainfall, from small to extreme. The majority of hourly precipitation simulated by WRF CTL is close to that in
observations, within the range of 1~10 mm/hr. In this range, future rainfall shows little increase compared to that under the current
climate, with even a slight decrease in the amount of light rainfall. The higher hourly rainfall at the high end of the distribution (>
mm/hr), although comprising only 0.5% of density in total events, shows a dramatic increase by a probability of 1.5 to 3 times
in frequency in the future warmer climate. Notably, the density in the high-end distribution is much higher in the station observation
than in CTL because the denominator for observed density, the total number of events, is significantly less in observation, although
the absolute number of high-intensity events is comparable or higher in WRF-CTL. In addition to a greater likelihood for the high
end of extreme rainfall occurring, a slight decrease in light rainfall is also evident, supporting previous findings from other
modeling studies (Cubasch et al., 2013; Easterling et al., 2000; Karl et al., 2006).

## 5      Extreme Temperature and Precipitation

In recent decades, there has been an increase in the number of hot extremes in Canada, particularly an increase of nighttime
temperature in summer as the global mean temperature rises. Both extreme cold and hot days greatly affect the economy, society,
and the daily lives of people. The changes in the high/low percentile values in the temperatures of WRF-PGW and CTL are used
to assess the future change in the extreme hot days in summer and cold temperatures in winter in western Canada. The 90th, 95th,
and 99th percentile of the daily maximum temperature for CTL, PGW, and their changes in summer are shown in Fig. 20(a). The
patterns of warming for the 90th to 99th percentiles are similar in summer and winter, respectively. The least warming occurs over
the central part of the domain where boreal forests are found, mostly within MRB with a magnitude of about 2.5℃. Over the
surrounding area, the warming is stronger with 4-5℃ for 90th percentile and 5-6 C for 99th percentile. The change in the 10th, 5th,





and 1st percentile of the daily minimum temperature in winter in WRF-PGW relative to WRF-CTL is shown in Fig. 20(b). In winter the strongest warming for the low percentile $T_{min}$ occurs in the eastern domain where the general warming is also stronger. The high percentile daily precipitation distribution in CTL and PGW simulation shows different geographical patterns for different percentiles in summer. The 90th, 95th,and 99th percentiles have typical values around 10,18,36 mm/day in the high precipitation region, respectively. Compared to CTL, the 90th percentile of daily precipitation in PGW in summer experiences little change in

the majority of the domain except for an increase (1.5-3 mm/day) in the Yukon and western MRB and a small decrease (-1.5 mm/day) in the southeastern domain, as shown in the first row in Fig. 21(a). The 95th percentile of PGW shows a more widespread increase of precipitation by 1-3 mm/day, compared to CTL, except for a small strip of decreased precipitation east of SRB, as shown in the second row of Fig. 21(a). The 99th percentile of daily summer precipitation shows a consistent increase of 6-9 mm/day and about a 15-30% increase across the domain. Extremely high percentile such as 99% is usually associated with synoptic weather

systems, for which the increase of precipitation is more uniform over the domain as it is proportional to vapor loading. The 90th percentile for summer precipitation, about 6-10 mm/day over the Canadian Prairies for CTL, can be associated with strong local thunderstorms, which will be strongly affected by boundary layer changes and lower atmospheric conditions in the future. Local water availability and partitioning between sensible and latent heat flux can change the convective inhibition and available convective potential energy, in turn affecting the convective precipitation. Therefore, there is large inhomogeneity of 90th

percentile summer precipitation over the domain compared to the lower and higher percentiles.

In winter the relative changes (PGW - CTL) in high percentile daily precipitation are similar for all the percentiles as shown in Fig. 21(b). The change in amount for each percentile follows the general pattern of high daily precipitation distribution in winter: it is concentrated along the coastal mountains in the west and in the Canadian Rockies. In terms of percentage increase, the largest increase is in the northern MRB and the northeastern domain, where precipitation is less than in other parts of the domain. The

pattern of the changes in extreme precipitation in winter follows the distribution of mean precipitation, indicating the increase mostly comes from more vapor loading in the atmosphere.

## 6    Discussion

The lack of observation presents challenges for regional climate modelling in several respects. The mountainous terrain and numerous lakes make interpolation of observation data to gridded data-sets difficult. Canada's meteorological observation network

is heavily concentrated in the southern part of the country and over the plains because of the higher population density and logistical factors. There are far fewer surface observation stations in the sparsely populated area in the north and over the mountainous regions. The sparse observation networks in the regions with low population density provide less reliable and representative observation data to develop and validate regional climate models in the region (Hofstra et al., 2009; Takhsha et al., 2017). As a result, the evaluation of model performance relative to a gridded observation dataset such as ANUSPLIN is less reliable in

mountainous and polar regions.

The cold region hydrological cycle and treatment of the snow cover in the land surface model component of RCMs also pose a great challenge to simulate the characteristics of surface temperature and hydrological processes in the region (Casati and Elía, 2014; Niu et al., 2011). For instance, cold region hydrometeorology is strongly affected by snow processes, which are, in turn, affected by fine-scale topography and wind transport. The representation of snow pack and cover in the mountainous region is a

challenging obstacle to overcome in realistically reproducing hydro-climatic conditions in the Canadian Rockies (Casati and Elía, 2014; McCrary et al., 2017; Niu et al., 2011). In our case, the near surface temperature in spring is highly sensitive to the



representation of spring snow cover as the snow-albedo feedback can amplify bias in the snow amount in winter to temperature bias in spring.

The convection-permitting high-resolution downscaling by WRF-CTL is in good agreement with CaPA in showing a precipitation pattern with a small wet bias in the northern part of the domain and mountainous region. Notably, however, the regions where WRF-CTL show wet biases also suffer from observation data scarcity or non-representativeness due to orographic precipitation.

In the PGW simulation, the Canadian Prairies, unlike other regions, show a slight decline or no change in total precipitation in summer, especially in moderate intensity precipitation compared to CTL. One reason there is little to no increase in precipitation in summer may be the decrease of relative humidity in the region, both in the PGW forcing and in the simulation. Dai et al. (2017) showed that a smaller increase of specific humidity than temperature rise can cause a decrease of relative humidity (see the PGW forcing in Fig. 2), as well as a much smaller increase of precipitation. The detailed mechanisms behind the suppression of summer

precipitation in the region compared to surrounding regions are currently under investigation.

The high intensity precipitation events are projected to increase by the end of the 21st century under RCP8.5,  as indicated by the notable increase of high intensity precipitation in the PDF of both MRB and SRB in WRF-PGW. Extreme precipitation is affected by both water-vapor loading in the atmosphere and changes in vertical velocity, size of storms, translation velocity of storms, etc. Large synoptic-scale storms tend to be affected by vapor loading more than by local-scale circulation, which is consistent with the

relative uniformity of the 99th percentile daily precipitation increase across the domain in all seasons. In contrast,  large regional differences are seen in the 90th percentile precipitation associated with lesser storms in summer. The hourly precipitation histogram shows a much larger increase in number for heavy precipitation events (about 300%) versus light precipitation (about 150%). Research is ongoing on changes of storm-related characteristics based on an objective storm tracking algorithm known as MODE-TD.

For many hydrological and agricultural applications, bias-correction of temperature and precipitation for RCM outputs often need to be reconciled with benchmarked parameters or criteria. Various bias correction methods have been used to bias-correct RCM output. Quantile mapping, in its various forms, tends to project simulated distribution onto the observed distribution and achieve observed mean and distribution. However, due to shifting in the distribution of hourly precipitation probability in WRF-PGW relative to WRF-CTL, quantile mapping for our simulation alters the precipitation change signal between the original WRF-PGW

and WRF-CTL. To preserve the climate change signal in bias-correction and to produce properly bias-corrected summer precipitation for future scenarios, the physical processes involved in the change need to be considered.

## 7   Summary

The 4-km WRF dynamical downscaling of the current and future (RCP8.5) climate provides valuable high-resolution regional climate data for applications in hydrology and climatic impact studies. High-resolution convection-permitting regional climate

simulations were conducted using WRF at 4-km grid-spacing for western Canada for the current climate (CTL, 2000-2015) and a high-end emission scenario RCP8.5 through the PGW approach. The WRF-CTL simulation is forced with ERA-Interim reanalysis at 6-h intervals on the boundary. The WRF-PGW's forcing is the same as that of CTL plus the climate change signals derived from an ensemble of 19 CMIP5 members from 2070-2100 and from 1976-2005. At a 4-km horizontal resolution, the convection in the model is explicitly resolved and the convective parameterization schemes are disabled.

The evaluation of WRF-CTL against the gridded observation dataset ANUSPLIN and reanalyses such as CaPA and NARR shows good agreement between WRF-CTL and the reference datasets in terms of geographical distribution, seasonal cycle, and interannual variation. For temperature bias, the largest bias occurs over the plains east of the Rockies in spring. In general, WRF-




CTL produces more precipitation than both ANUSPLIN and CaPA, especially in the northern part of domain where there are few observations and over terrains where most observation sites are at a lower elevation and less representative than desired. The precipitation bias of WRF-CTL against CaPA is less than that versus ANUSPLIN, which has been shown to be too dry in the northern domain and SRB (Wong et al., 2016).

In a future warming scenario, WRF-PGW shows substantial warming across western Canada under the RCP.8.5 emission scenario,
though the warming is slightly less than that from the CMIP5 ensemble projection. The warming is stronger in the cold season, especially over the northeast polar region in winter and over the Canadian Prairies in spring. While precipitation changes in PGW over CTL vary with the seasons, in both basins, more increases occur in spring and late fall, whereas precipitation in SRB in summer either shows no increase (remains at zero) or decreases. The smallest change in precipitation occurs in summer, when parts of the Prairies receive less precipitation in PGW than in CTL. With almost no increase in summer precipitation and much
larger evapotranspiration in PGW than in CTL, the Canadian Prairies will experience water availability challenges during the growing season. With the large temperature increase and potential increased evaporation, the small increase/decrease of summer precipitation indicates that the Canadian Prairies and the southern Boreal Forest biomes will see a slight decline in effective precipitation in summer, which will likely have a significant impact on soil moisture for farming and forest fire occurrences.

As the warmer atmosphere holds more water vapor, WRF-PGW shows an increase of precipitation over WRF-CTL. Severe
precipitation events increase more than moderate and light precipitation events as the distribution of precipitation events shifts toward more higher intensity events in all seasons except summer. In summer, light to moderate precipitation (5-10 mm/3hr) in WRF-PGW actually decreases compared to WRF-CTL in both MRB and SRB. The increase of precipitation in cold seasons is larger in terms of percentage in the northeast where greater warming is expected. Due to this shift in precipitation intensity to the higher end in the PGW simulation, the seemingly moderate increase in total precipitation in summer for both basins may not reflect
the real change in flooding risk and water availability for agriculture because the frequency of extreme precipitation events increases disproportionately. The shift in the precipitation intensity distribution in WRF-PGW also poses challenges, for bias-correction relies on fitting an observed distribution.

In summary, the high-resolution convection-permitting WRF simulations are shown to reproduce the general characteristics of the regional climate in western Canada. The model results provide bountiful opportunities for not only atmospheric and climate
scientists interested in local-regional scale meteorological phenomena and dynamics and circulation changes under global warming, but also stakeholders in hydrology and agriculture who need high-resolution climate information and detailed global warming projections for western Canada.

**Author contributions**

YL designed the experiments and CL, ZZ, SK, ZL performed the simulations. ZL, ZZ conducted the analysis and prepared the
figures with the helps from YL, LS and PX. YL and ZL prepared the manuscript with contributions from all co-authors.

**Acknowledgments**

The authors gratefully acknowledge the support from the Changing Cold Regions Network (CCRN) funded by the Natural Science and Engineering Research Council of Canada (NSERC), as well as the Global Water Future project and Global Institute of Water Security at University of Saskatchewan. Yanping Li acknowledge the support from NSERC Discovery Grant.





The authors declare that they have no conflict of interest.

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

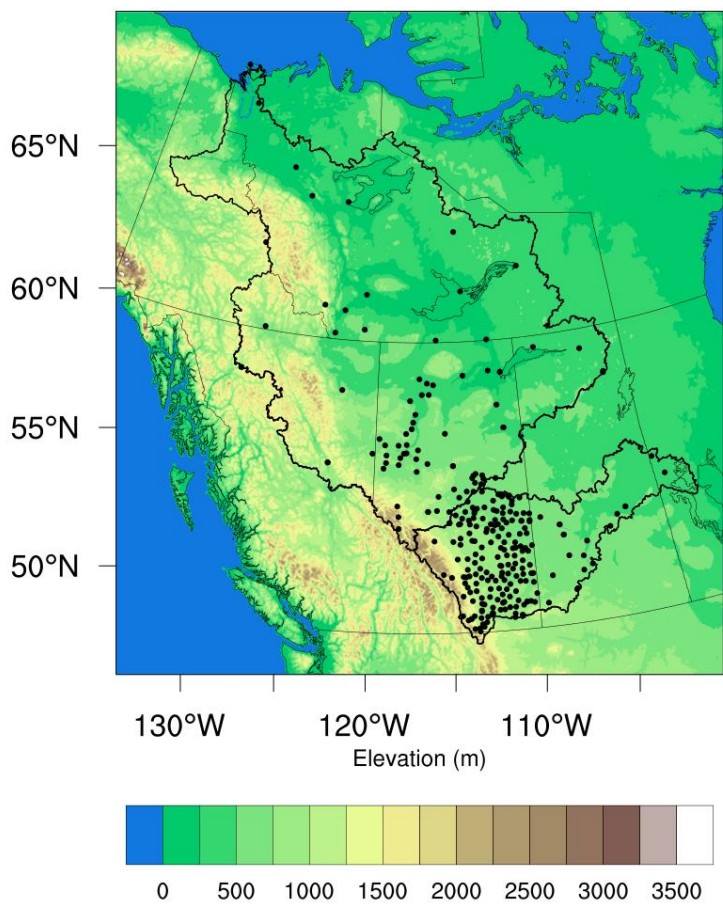

**Figure 1: The domain of WRF simulation. The black dots indicate the observation stations used in the evaluation of the simulations.**





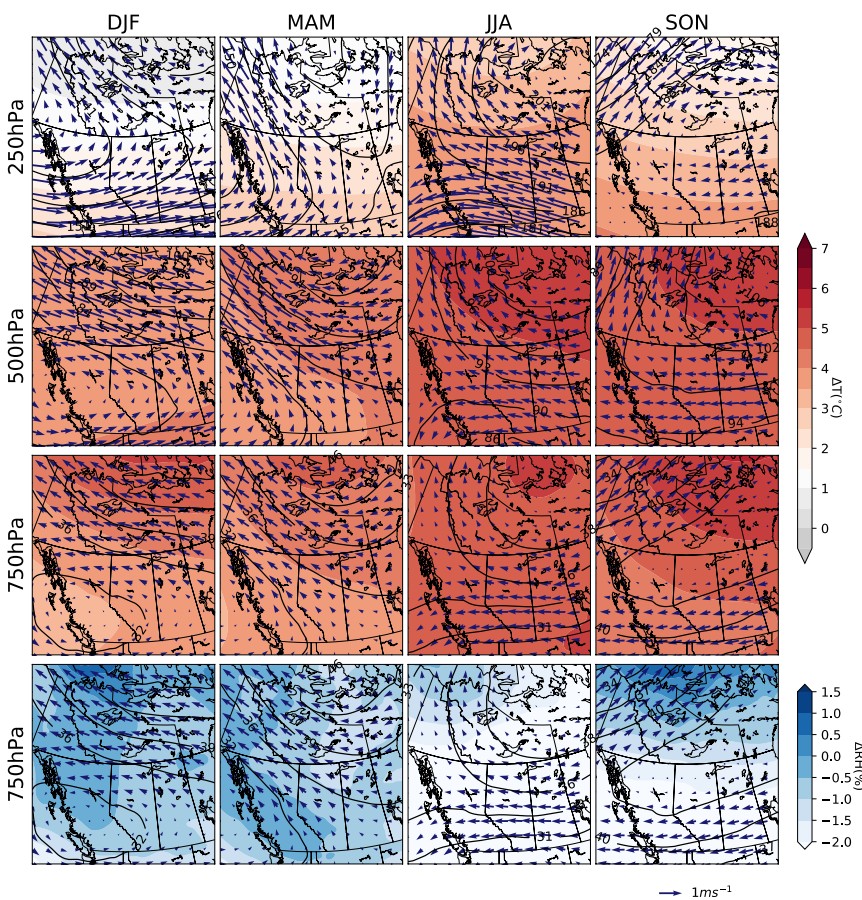

**Figure 2: The climate change signal in the PGW forcing derived from 19-member CMIP5 ensemble. The contours are the changes in geopotential height relative to current climate. The shadings are changes in temperature or moisture at each pressure level. The wind vectors denote the change in the mean wind at each level.**





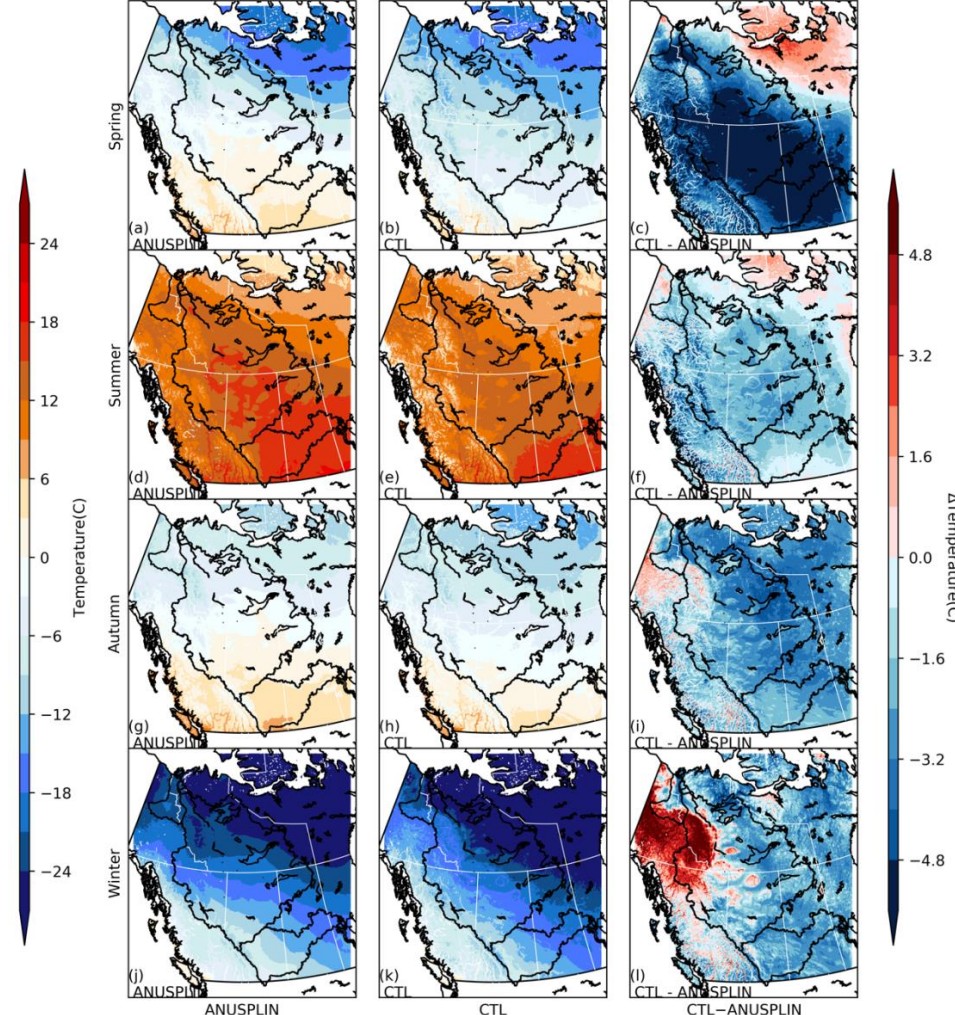

**Figure 3: The daily mean temperature for spring (MAM), summer (JJA), and autumn (SON), and winter (DJF) from 2000 to 2015 of ANUSPLIN (left column), WRF-CTL (middle column) and the difference (CTL - ANUSPLIN, right column).**



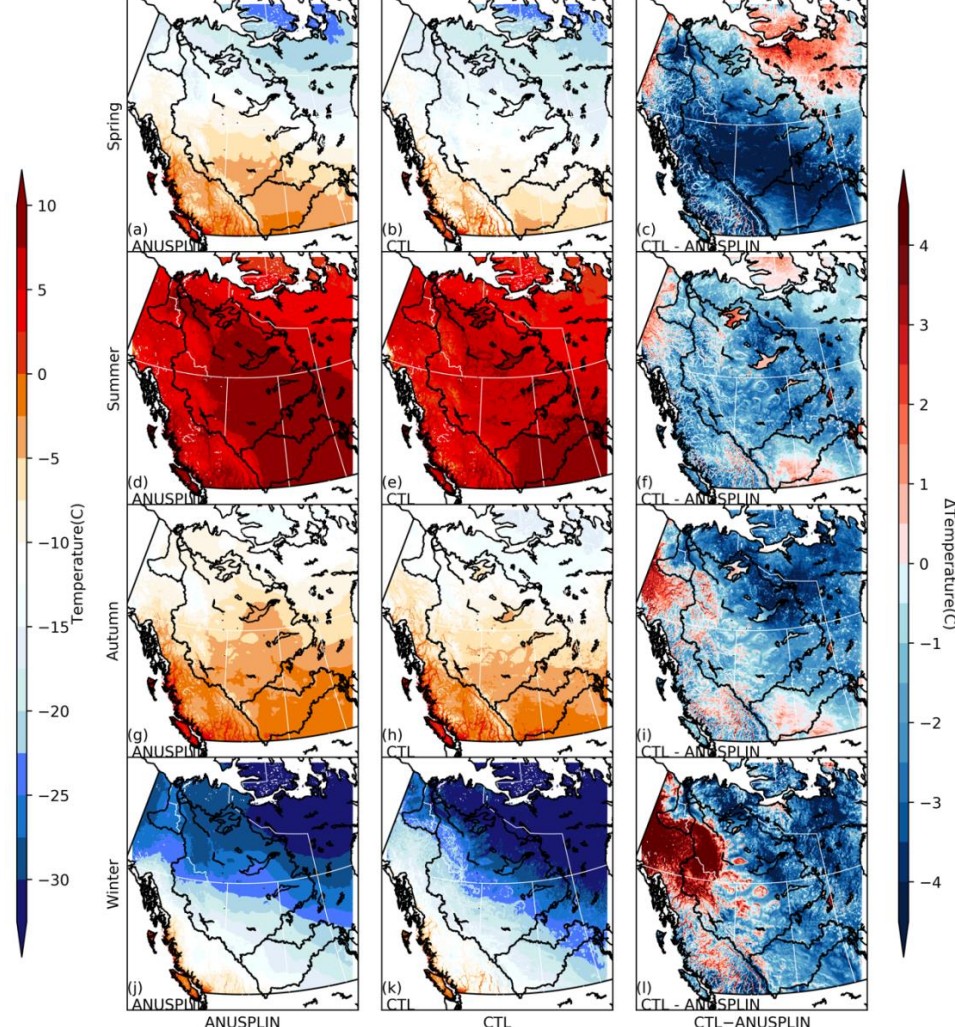

**Figure 4: The seasonal averaged daily minimum temperature from ANUSPLIN (left column), WRF-CTL (middle column) and the difference (CTL - ANUSPLIN, right column).**





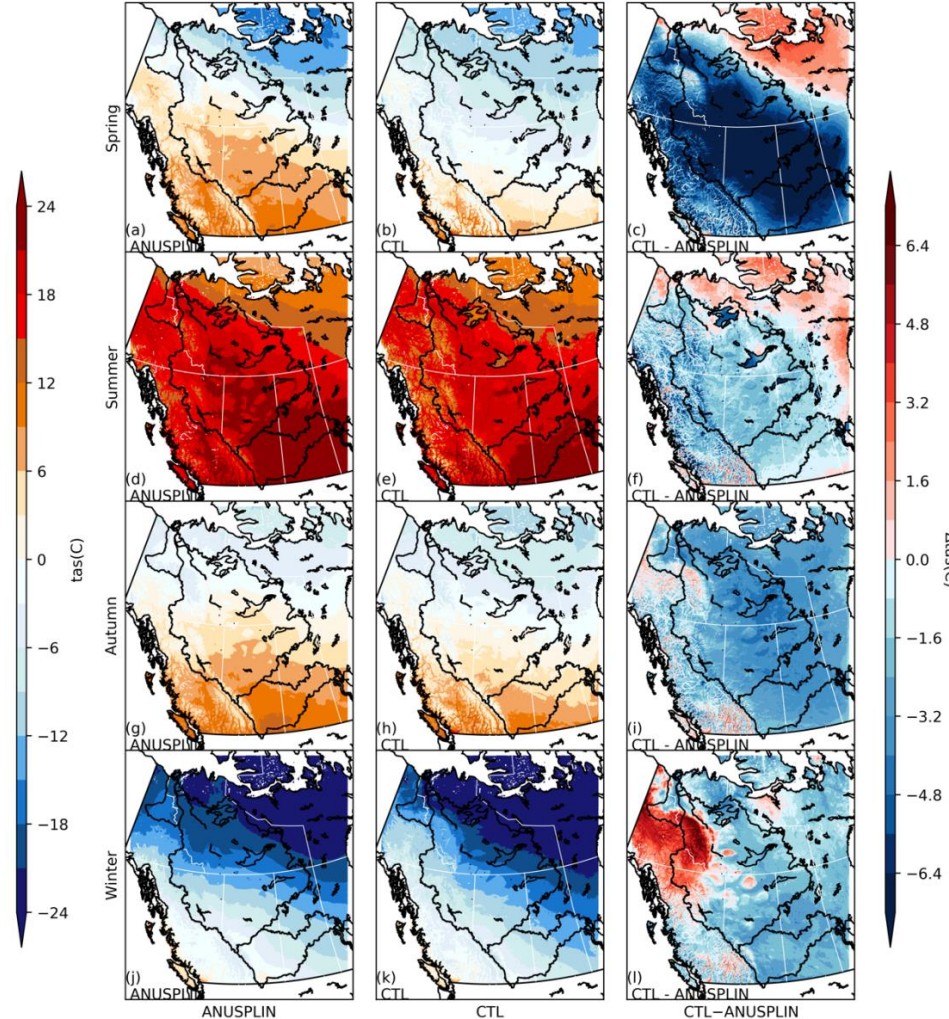

**Figure 5: The seasonal averaged daily maximum temperature from ANUSPLIN (left column), WRF-CTL (middle column) and the difference (CTL - ANUSPLIN, right column)**





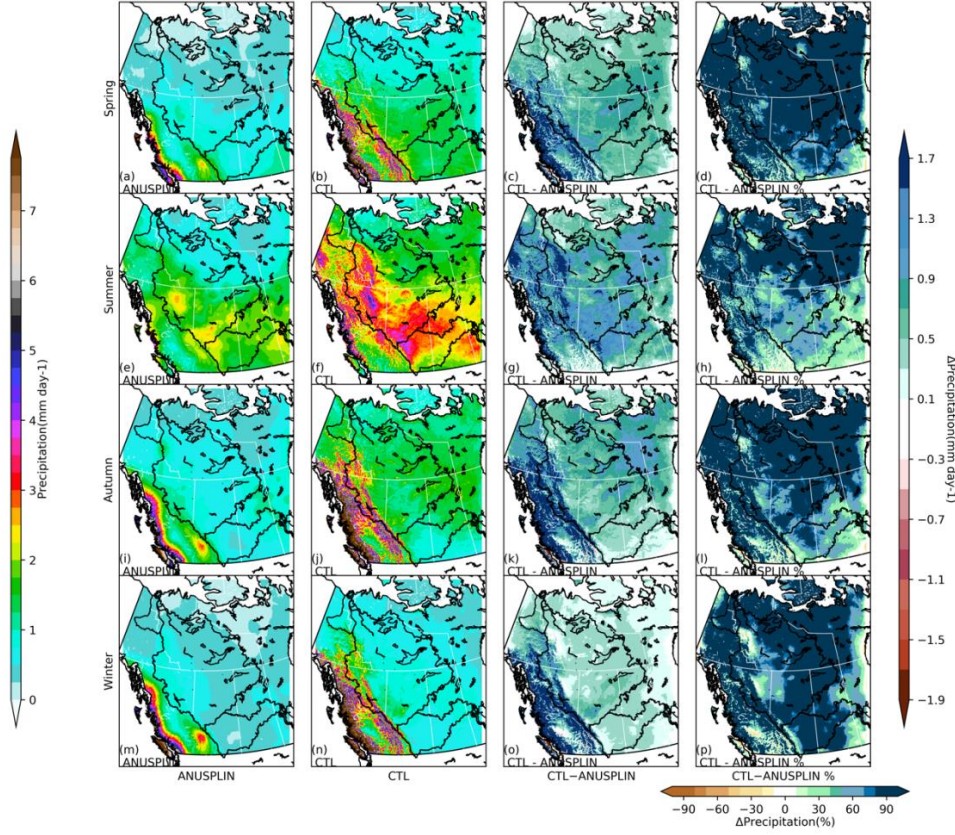

**Figure 6: The daily mean precipitation from ANUSPLIN(1st column) and WRF-CTL(2nd column), and their absolute (3rd column) and relative differences in percentage (4th column).**





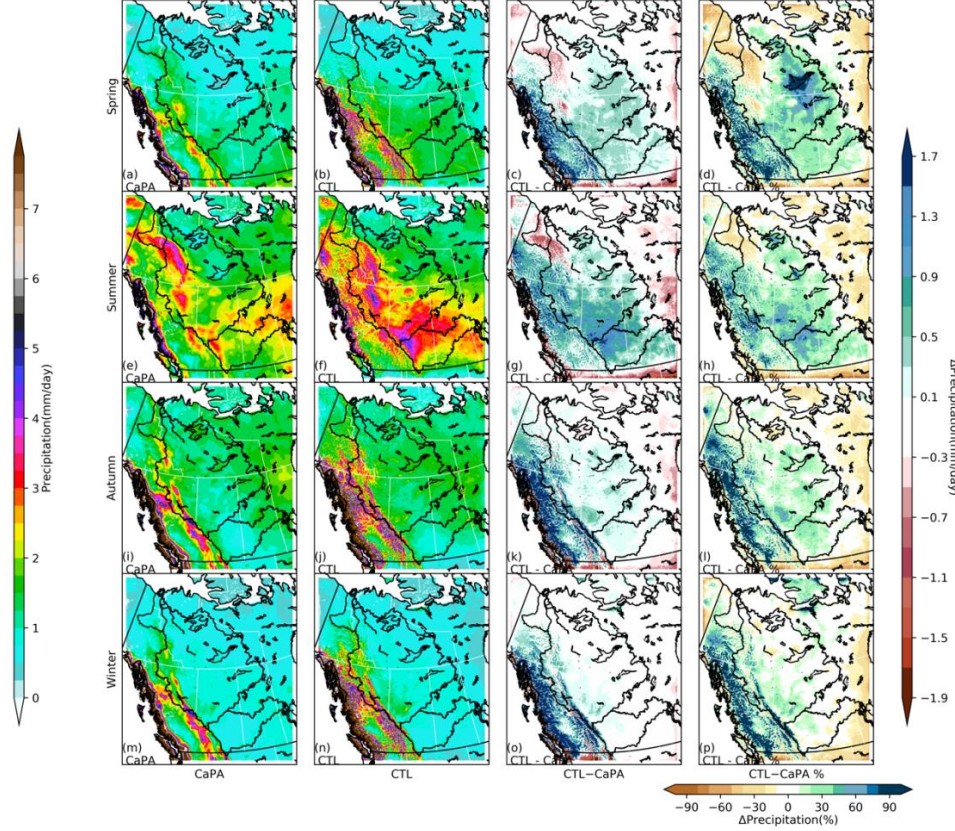

**Figure 7: The daily mean precipitation from CaPA (1st column) and WRF-CTL(2st column), and their absolute (3rd column) and relative differences in percentage (4th column).**





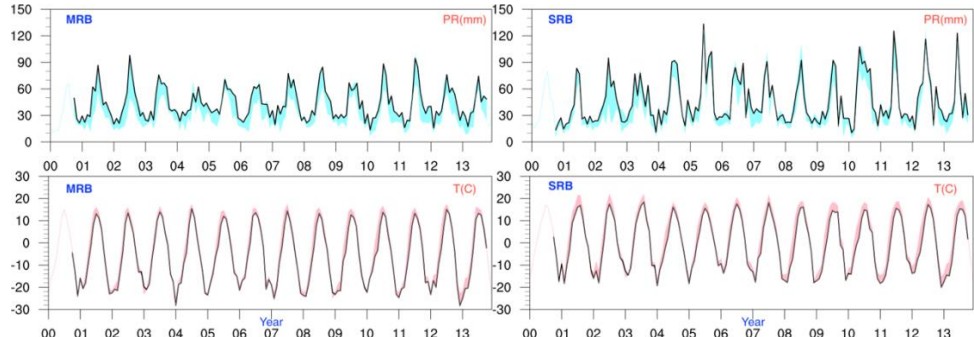

5    **Figure 8: The monthly mean precipitation/temperature averaged over the Mackenzie River Basin (left) and Saskatchewan River Basin (right) from 2000 to 2015 from WRF-CTL (black curve) and an ensemble of observation of precipitation (aqua shading) and temperature(pink shading).**

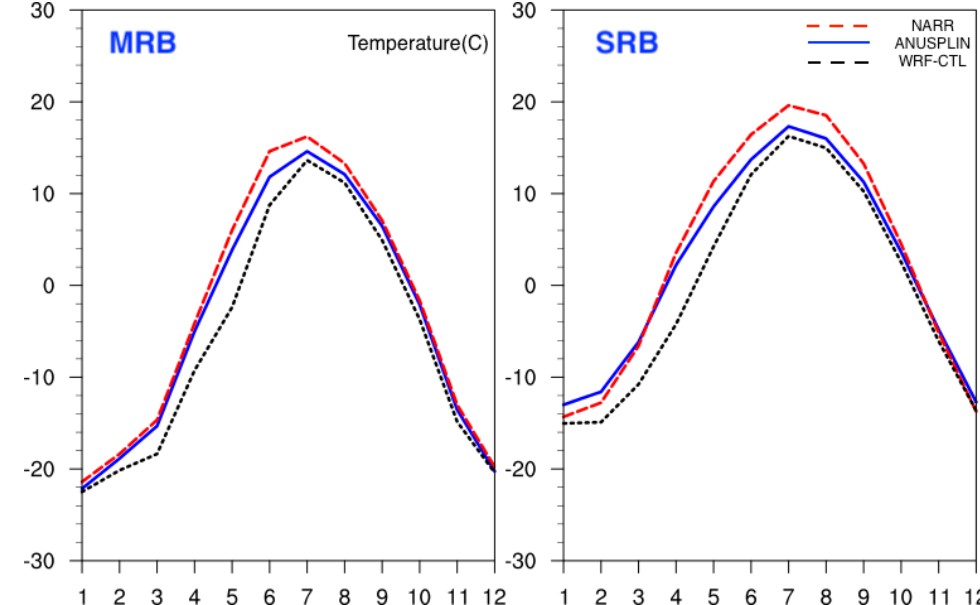

10   **Figure 9:  The mean annual cycle for WRF-CTL (black), NARR (red), and ANUSPLIN (blue) over the Mackenzie River Basin (left) and Saskatchewan River Basin (right). Monthly basin averaged precipitation over Saskatchewan River Basin from WRF-CTL, WRF-CONUS control run and the ensemble of observation datasets (NARR, ANUSPLIN, CaPA).**



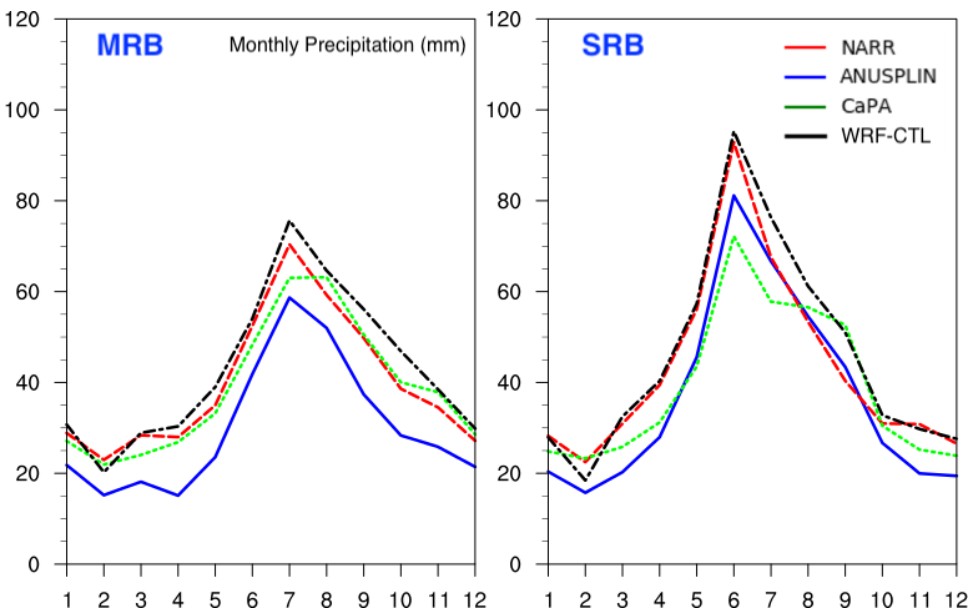

**Figure 10: The mean annual cycle of monthly precipitation for WRF-CTL (black), NARR (red), and ANUSPLIN (blue)
over the Mackenzie River Basin (left) and Saskatchewan River Basin (right).**

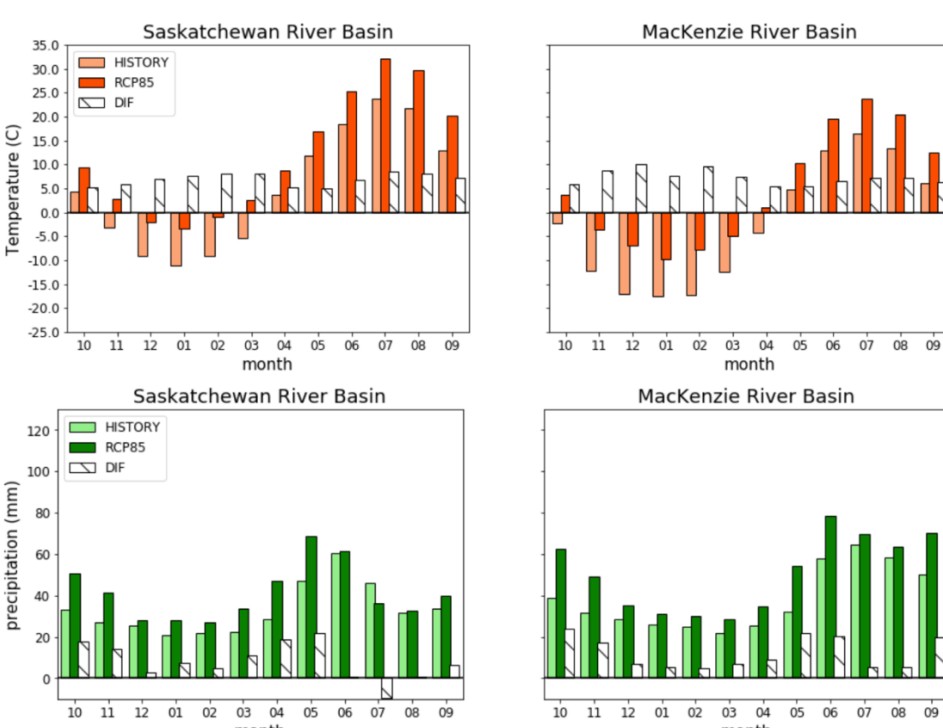

**Figure 11: Annual cycle of temperature and precipitation projected by CMIP5 ensemble. The orange bars indicate the
basin average temperature of current climate (1976-2005). The red bars represent the basin average temperature at the**





5    **end 21st century under RCP8.5(2076-2100). The white bars denote the change in temperature at the end of century relative to the current climate. The dark green bars indicate the basin average precipitation of current climate. The shallow green bars represent the basin average precipitation at the end 21st century under RCP8.5. The white bars denote the change in precipitation.**

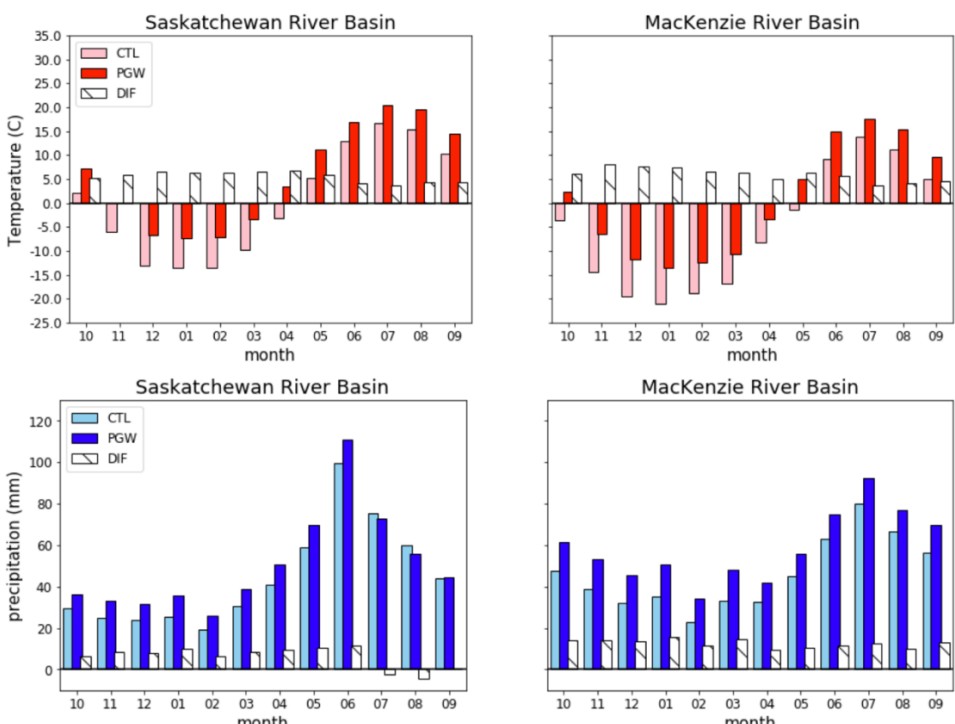

10    **Figure 12: Annual cycle of temperature and precipitation projected by WRF-PGW (2001-2015). The pink bars indicate the basin average temperature of current climate. The red bars represent the basin average temperature at the end 21st century under RCP8.5. The white bars denote the change in temperature at the end of century relative to the current climate. The dark blue bars indicate the basin average precipitation of current climate. The shallow blue bars represent the basin average precipitation at the end 21st century under RCP8.5. The white bars denote the change in precipitation.**





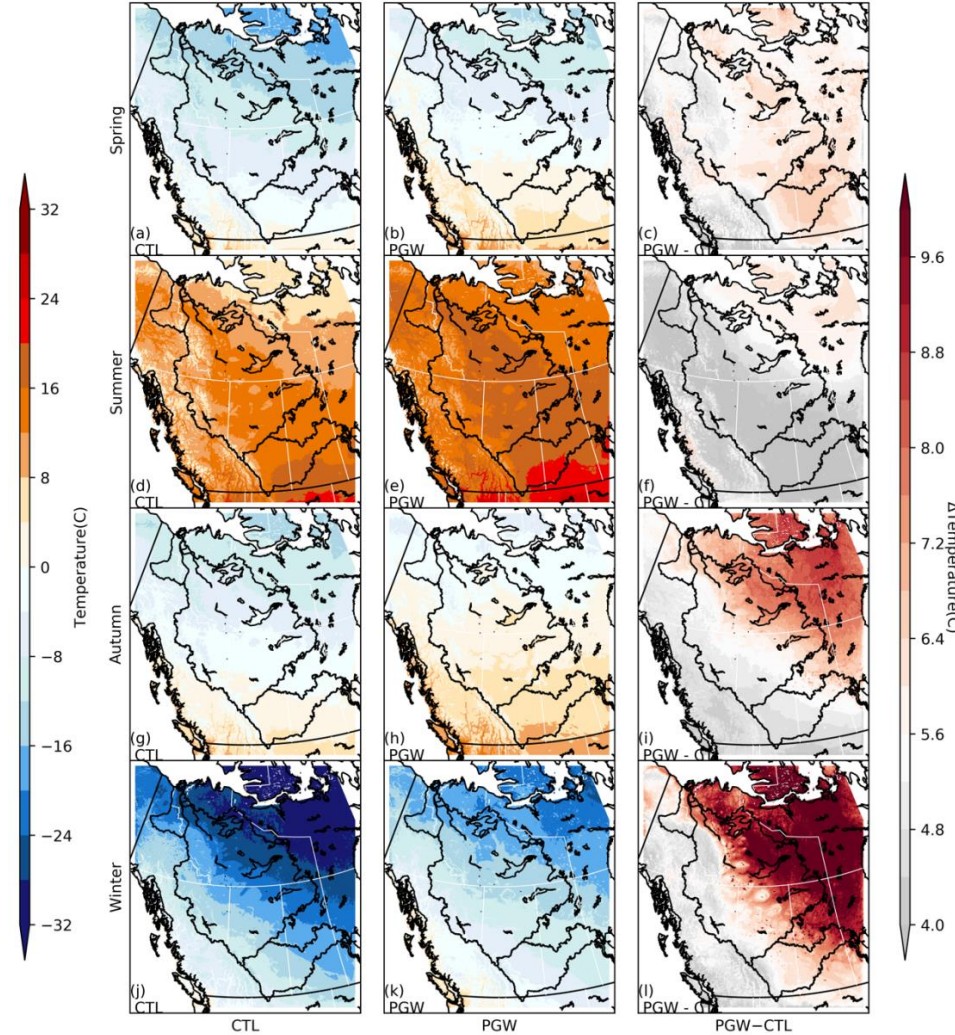

**Figure 13: Daily mean temperature from WRF-CTL (left) and WRF-PGW (middle) and the difference (PGW - CTL, right)**





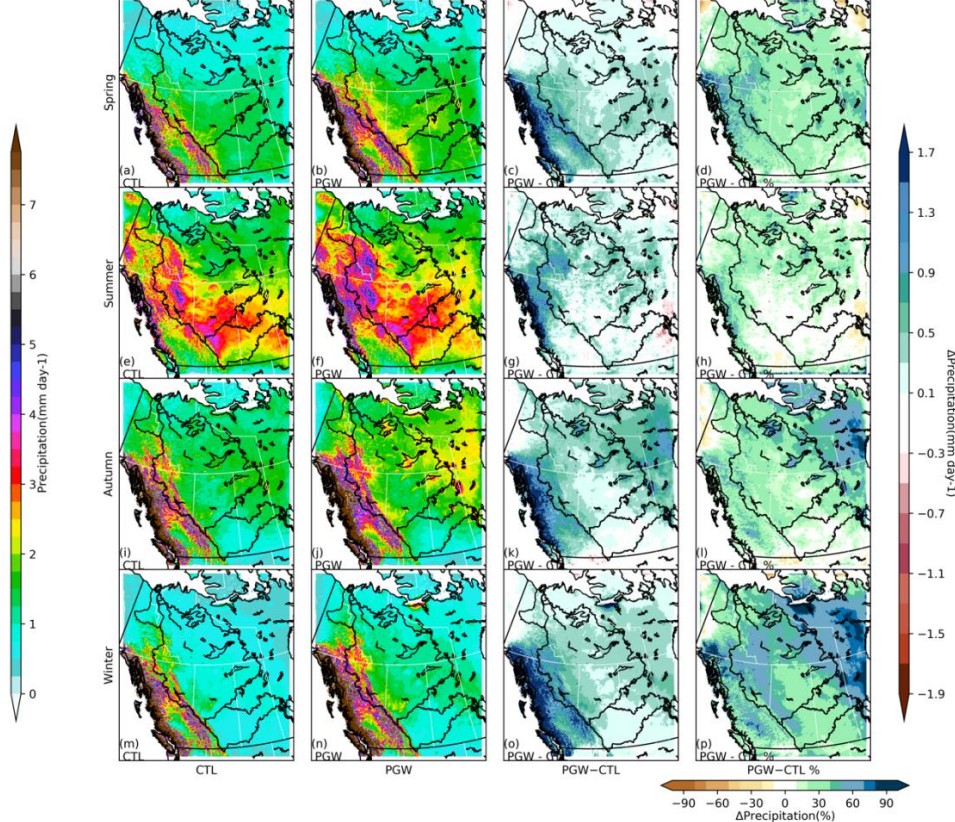

**Figure 14: Daily mean precipitation from WRF-CTL (top) and WRF-PGW (middle), the difference (PGW - CTL, second to the right), and percentage difference over CTL (right).**





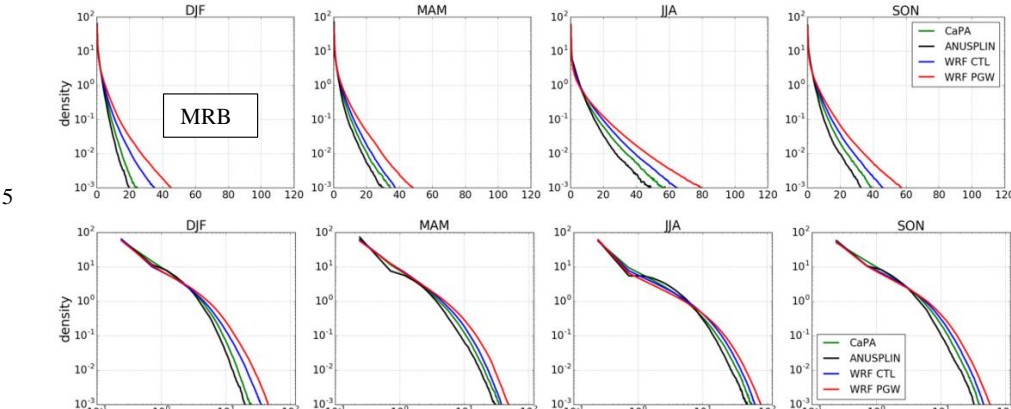

**Figure 15: Daily precipitation probability density function in Mackenzie River Basin for WRF (CTL, PGW) and CaPA, ANUSPLIN with linear-log, log-log axes for density (y) and amount (x).**

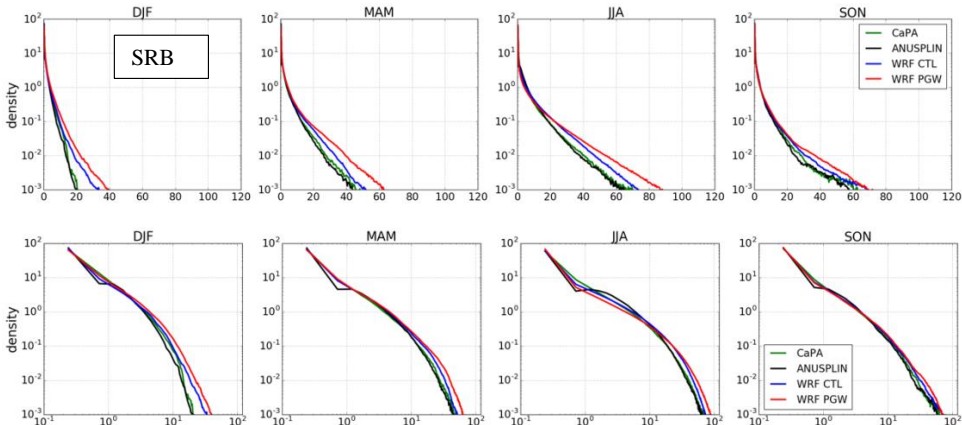

**Figure 16: Daily precipitation probability density function in Saskatchewan River Basin for WRF (CTL, PGW) and CaPA, ANUSPLIN with linear-log, log-log axes for density (y) and amount (x).**




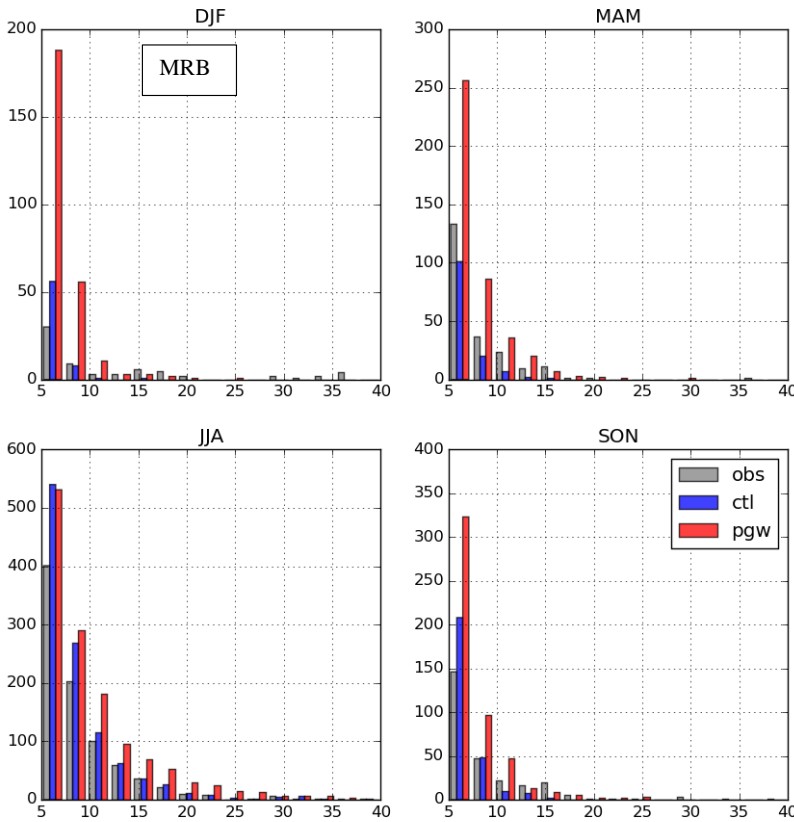

**Figure 17: Three hourly precipitation distribution from station observation in MRB and those corresponding to WRF-CTL, WRF-PGW simulations.**





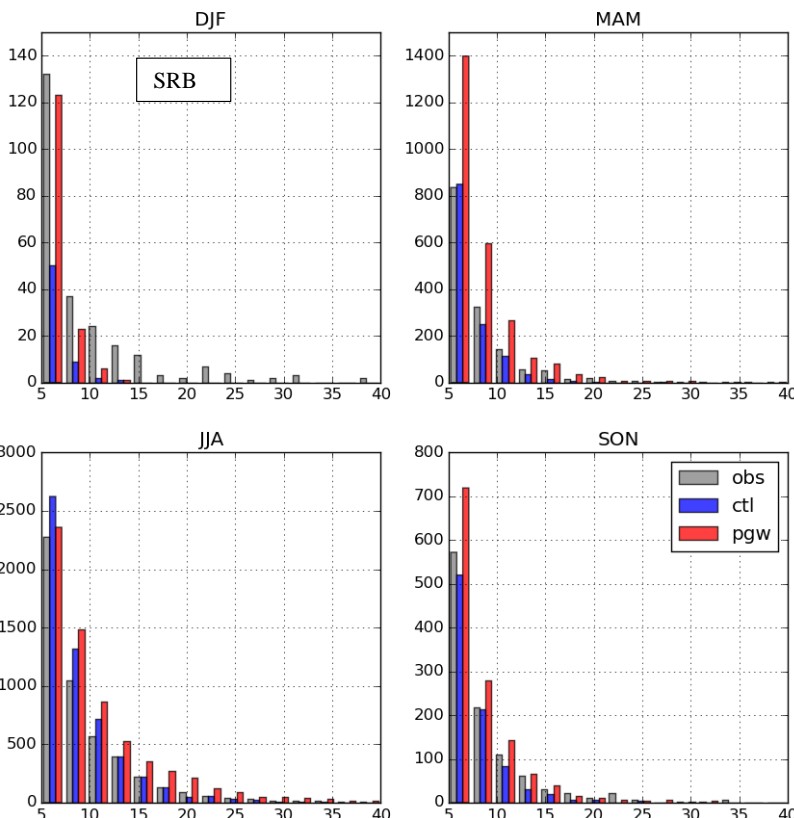

**Figure 18: Three hourly precipitation distribution from station observation in SRB and those corresponding to WRF-CTL, WRF-PGW simulations.**





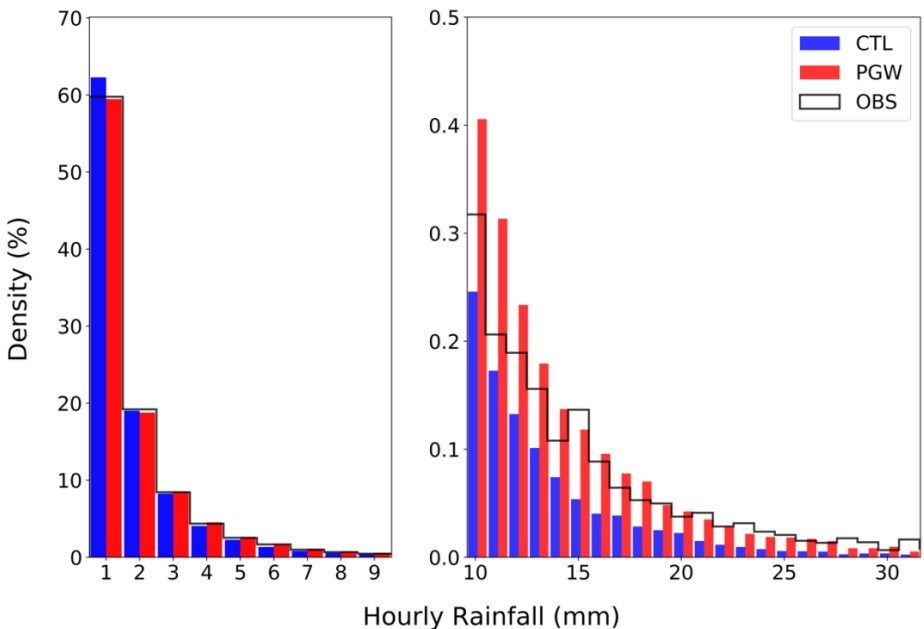

**Figure 19: Hourly extreme precipitation frequency density over western Canada from station observation, WRF-CTL and PGW. The bottom panel shows the ratio between PGW and CTL for events with different intensities.**





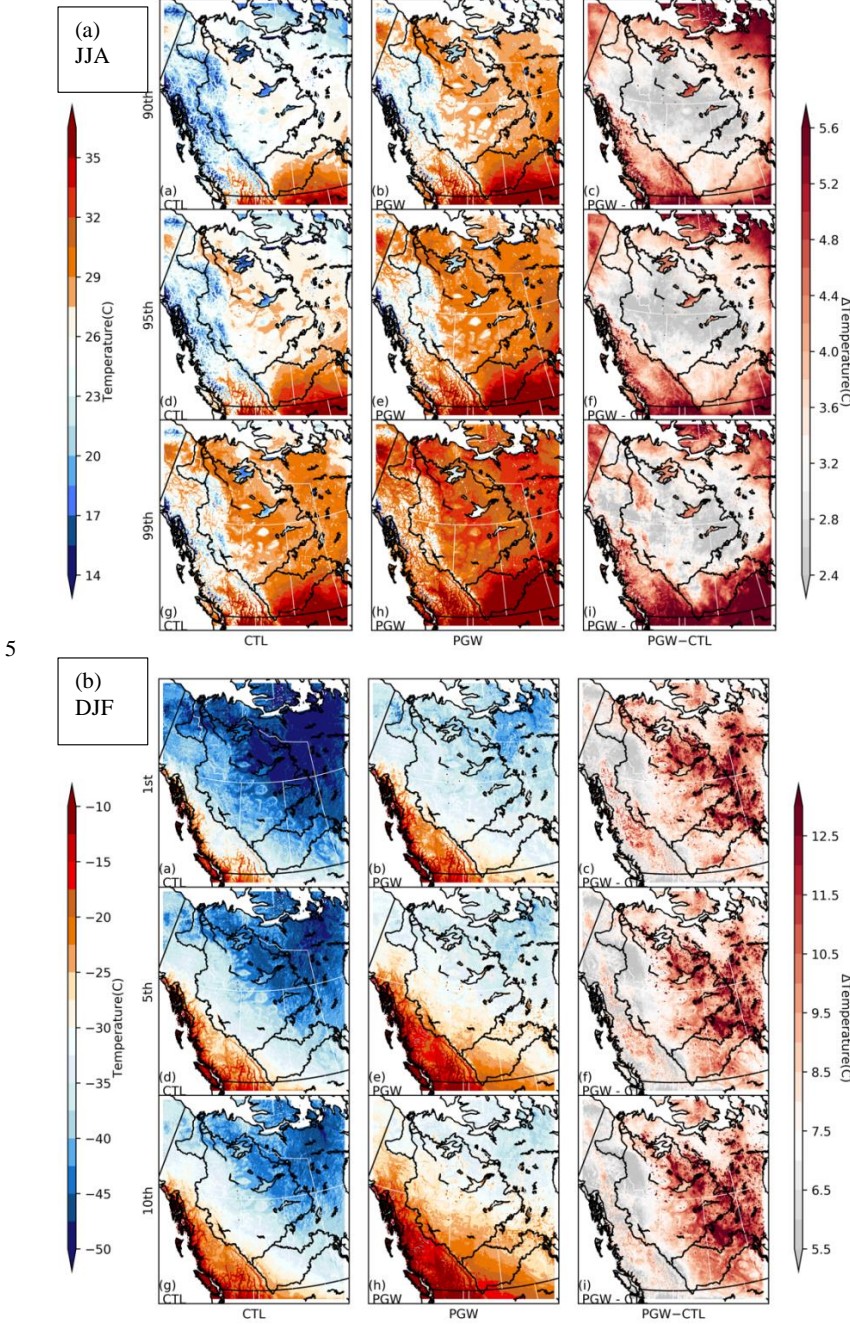

**Figure 20:** ( a) Extreme statistics of daily maximum temperature in summer WRF-CTL vs WRF-PGW, from top to bottom: 90th, 95th, and 99th percentile. (b) Extreme statistics of daily minimum temperature in winter WRF-CTL vs WRF-PGW, from top to bottom: 1st, 5th, and 10th percentile.





**Figure 21: Extreme statistics of daily precipitation in summer (a) and winter(b) for WRF-CTL vs WRF-PGW, from top to bottom: 90th, 95th, and 99th percentile.**