# Peer review of "High-Resolution Regional Climate Modeling and Projection over Western Canada using a Weather Research Forecasting Model with a Pseudo-Global Warming Approach"

_Hydrology and Earth System Sciences, 2019_

## Referee Comment (RC1) · Anonymous Referee #1 · 7 Jun 2019

(2019) Summary

The authors perform two sets of simulations for the western Canadian domain via a dynamical downscaling approach. First, a control run (CTL) of the historical climate from October 1, 2000 to 30 September 2015 forced at the boundaries using 6-hourly 0.7 degree ERA-Interim reanalysis data was carried out. Subsequently, a pseudo-global warming (PGW) run of future climate following the RCP8.5 pathway was produced

for the same period as the CTL run but using temporally averaged fields from an ensemble of CMIP5 climate model simulations (2071-2100 relative to 1976-2005) and adding those to the initial and boundary conditions of the CTL experiment. The analysis is conducted at the convection-permitting scale (4-km) within the Weather Research Forecasting (WRF) model. The authors validate the CTL run against gridded observations in this region of Canada and found close agreement with reality although wet biases in precipitation are found as well as cold bias in spring air temperature particularly over the Rocky Mountains and towards the west coast. In terms of future climate, the PGW simulation shows more warming than CTL while precipitation changes are seasonally dependent. Increases are expected in the spring and fall while no change or decreases are anticipated in the summer months. The paper is generally well written and the authors made substantial efforts to verify the simulations. Please find below my major and minor comments for consideration.

Major comments

Objective of the study: performing a fine scale simulation (CTL and PGW) which captures convective cells at the 4km resolution shouldn't be the only objective of this paper. It is obvious that most of Canada has inevitably limited ground measurements which tend to not capture the effects of mountainous topography in areas such as the Rocky Mountains which are headwaters for most of the rivers flowing in either side of the Cordillera. As judged by the extensive verification of the simulations you carry out in the paper, it is possible that your study could explore the capability of the current generation of mesoscale atmospheric models such as WRF to reproduce precipitation and temperature features which are important for hydrology and water resources applications in western Canada. Simply, you have to also stress the need for improved forcing products given the data paucity in many regions of Canada.

Pseudo-Global Warming (PGW): what is the rationale behind employing the PGW downscaling approach instead of the continuous downscaling method often used in region experiments such as NARCAPP and CORDEX? You state in Section 2.2 that

'The second simulation was a climate perturbation or sensitivity experiment following the Pseudo-Global Warming (PGW) approach used in Colorado-Headwaters work (Rasmussen et al., 2014, 2011). Climate projections from GCMs introduce large uncertainties because of the substantial inter-model variability among GCMs (Deser et al., 2012; Mearns et al., 2013), which can obscure the climate change response due to global warming. Using the PGW approach, rather than the intermodel variability, can isolate radiative forcing and its associated circulation as the sole reason for the regional climate response'. What the many disadvantages of the PGW method? See for example:

Sato, T., F. Kimura and A. Kitoh, 2006: Projection of global warming onto regional precipitation over Mongolia using a regional climate model, Journal of Hydrology, Volume 333, Issue 1, 30 January 2007, Pages 144-154.

Vasubandhu, M. and M. Kanamitsu, 2004: Anomaly Nesting: A Methodology to Downscale Seasonal Climate Simulations from AGCMs. Journal of Climate, vol. 17, Issue 17, pp.3249-3262

Added value of convection permitting downscaling: the overarching aim of this study is to provide high resolution modelling of mesoscale meteorology phenomena over the western Canadian region. However, the authors haven't paid attention to the 'added value' of running WRF at 4km. For example, why should one trust the precipitation simulated by WRF at 4km compared to a low resolution output such as that of the Community Earth System Model (CESM)? This could have been the main stem of this study but it is not addressed here. I urge the authors to address this issue using any other available climate model products or observational products. For example, a validation could be made of WRF-resolved convective precipitation and other available products.

Validation of WRF CTL against observations: I am surprised by the agreement between the WRF CTL simulation and the observed data sets over the domain. The WRF

simulations in this study should inherit the biases from its forcing (ERA-interim) nudging was not involved in the downscaling procedure. In such cases, some form of bias correction is often applied to the WRF output. A related question is how good are the ANUSPLIN, NARR (~32km) and CaPA estimates relative to station observations over the study domain? These products are issued at ~10 km, so you upscaled 4km WRF to 10 km and downscaled NARR to 10 km for direct comparison? I suppose several other station observations exist especially over the poorly gauged rocky mountain chains which could give us a better insight on the performance of WRF CTL in high elevation areas. Are mountain weather processes (especially in winter) well captured in the ANUSPLIN and CaPA data products? If not, this is a limiting factor to the verification you have shown in the paper.

The coarse (~80 km) grid spacing of ERA-interim: The region of Canada where you made the WRF simulations is topographically complex and challenging to model with a coarse DEM and grid spacing of ERA-interim to effectively capture the orographic effects in air temperature and precipitation. Such a coarse resolution could underestimate the complexity of topography in western Canada. Did you check to ensure that the highest peak of Rocky Mountains used in the WRF DEM is comparable to that in ERA-interim? This could also explain a lot of the biases you found in this study.

Sensitivity of various schemes and selection methods: the feeling I get after reading through this study and that of Liu et al, 2017 is that most of the schemes used in the model set up over the CONUS were simply repeated for the Canadian domain. This could be probabilistic to model water balance and energy exchanges in more snow dominated high latitudes using the same schemes as over the US. The follow up question is how you went about selecting the combination of parameterization schemes, planetary boundary layer schemes, and radiative transfer schemes for Canada north of 50o? Some results from the diagnostic test runs should be included as supplementary material. This is a study which involved model set ups and numerical experiments, thus the authors should elaborate on their modelling strategy by providing these details.

Liu, C., Ikeda, K., Rasmussen, R., Barlage, M., Newman, A. J., Prein, A. F., Chen, F., Chen, L., Clark, M., Dai, A., Dudhia, J., Eidhammer, T., Gochis, D., Gutmann, E., Kurkute, S., Li, Y., Thompson, G. and Yates, D.: Continental-Scale ConvectionPermitting Modeling of the Current and Future Climate of North America, Climate Dynamics, 49(1), 71–95, doi:10.1007/s00382- 016-3327-9, 2017.

Suitability for impact studies: 'For many hydrological and agricultural applications, bias-correction of temperature and precipitation for RCM outputs often need to be reconciled with benchmarked parameters or criteria' Are you recommending some form of bias correction be applied to the data before using it for impact analysis? The narrative throughout the paper is that WRF compared well with observations if not of some wet and cold biases over the mountain areas and along the coast. Based on your analyses, would you recommend that bias correction be applied to the WRF data for the whole domain or only over the regions mentioned above? Figure 3-11 indicate that WRF is quite biased compared to observations if you look closely at the mean error maps. A 4 degree cold bias has serious implications for snow hydrology.

Figure 8: taking an ensemble spread of ANUSPLIN, NARR and CaPA to compare WRF against isn't correct. I see this in some studies in the scientific literature but it's ill-informed. These observations are not climate model outputs that respond to the same forcing and can be perturbed to generate an ensemble. The purpose of the verification exercise is also to assess the skill of various observed products and where WRF stands in comparison. Simply computing the spread of the three observational data sets and comparing WRF against doesn't make any sense. Consider comparing WRF against each of these products.

Minor comments

- It is not a wise thing to do by not providing continuous line numbers for the manuscript. I see this in a lot of manuscripts often submitted by newcomers to the field. - Abstract: the authors should consider reducing the length of the abstract. What do you mean

by 'Due to this shift in precipitation intensity to the higher end in the PGW simulation, the seemingly moderate increase in the total amount of precipitation in summer for both the Mackenzie and Saskatchewan river basins may not reflect the real change in flooding risk and water availability for agriculture'? This kind of ambiguous statement isn't suitable for inclusion in the abstract. Consider revising the abstract in simple terms without leaving the reader having to read the body of the manuscript in order to understand the abstract. - Change 'and for studying climate impacts in hydrology, agriculture, and ecosystems' to 'and for studying climate impacts on hydrology, agriculture, and ecosystems' - 'The change in probability distribution of precipitation intensity also calls for innovative bias-correction methods to be developed for the application of the dataset when bias-correction is required'. This isn't the crux of the paper and shouldn't be the closing statement of your abstract. Rather you should highlight the importance of high resolution modelling compared to other mesoscale downscaling approaches which still parameterize convection.

Figures

- The mean error maps have a delta sign beside the colorkey. What does it represent? Some readers may take it to mean a change a precipitation or temperature between two periods.

- Write degree Celsius as (oC) in all figures.

- Figure 4: the colorkey to the left is not informative. You need a diverging color ramp with 0 at the centre to represent the 'white' color break. It is not possible to interpret the figure in its current form.

- Figure 6 and 7: the colorbreak labels are not informative. Use equal breaks for both wet and dry

- Figure 16: why a log and not a linear scale for daily precipitation?

[Figure]

201, 2019.

---

## Referee Comment (RC2) · Anonymous Referee #2 · 29 Jun 2019

This manuscript presents convection-permitting regional climate simulations over western Canada. Specifically, two simulations with the 4-km-resolution WRF model are performed, comprised of a reanalysis-based historical climate simulation and a future end-of-the-century climate simulation using the pseudo-global warming (PGW) method. The validation of the historical simulation shows reasonable capability of the convection-permitting model at reproducing the observed seasonal climatological patterns of near-surface temperature and precipitation, but the widespread cold and wet biases are present. The PGW simulation indicates an increase of seasonal precipitation over most areas across all seasons, with a shifting in precipitation intensity to the higher end. Overall, the manuscript is clearly written and in good shape, but some modifications/clarifications are needed for the acceptance for publication.

1. The writing is too wordy and also there are too many figures. Here are a few examples:

1) The description of the CMIP5-derived perturbation on page 5 is too detailed. I think the authors only need to present the major features of the dynamical and thermodynamical changes.

2) Figures 3-5: I don't see the need for presenting both the daily mean and the daily maximum/minimum temperature because the mean is just an average of the maximum and minimum.

3) Figures 6-7 can be merged into one. Alternatively, just remove Figure 6 if the authors don't trust ANUSPLIN data.

4) Figures 20-21: Because of the great similarity in warming pattern between the three percentiles, there is no need to show all of the corresponding plots.

2. There are many inaccurate statements, as well as some grammatical errors and typos. Here are just a few examples in the abstract (page 1). 1) L18-19: explicitly resolving cumulus plumes. This is not true. To resolve individual convective elements a sub-kilometer grid spacing is necessary. 2) L19-21: How can you conclude that the simulation agrees with observations in terms of the geographical distribution of cold bias? Logically, this is wrong. 3) L24: "the PGW simulation shows more warming than CTL". The authors may want to say "the PGW simulation shows significant warming relative to CTL".

3. Section 4.1. For a fair comparison between WRF downscaling and CMIP5 projection, the temperature and precipitation changes for CMIP5 should be computed as the difference between the 1976-2005 average and the 2071-2100 average, consistent

with the climate perturbation used for PGW (i.e., Eq.1).

4. I'd like to suggest presenting the temperature and precipitation changes over the whole domain (i.e., Figure13-14) first, followed by describing the sub-domain results (Figs. 11-12).

5. Add CMIP5 projected changes in Figures 13-14.

6. Page 5, L8: add "the change of cloud population (Rasmussen et al. 2018)".

---

## Author Comment (AC1) · 13 Sep 2019

We thank the reviewer for kindly reviewing our manuscript. Many aspects of the paper have been improved by addressing to the points raised by the reviewer. We hereby sincerely thank the great effort by the reviewer in conducting a thorough review of the draft.

**1  Major comments**

*Objective of the study: performing a fine scale simulation (CTL and PGW) which captures convective cells at the 4km resolution shouldn't be the only objective of this paper. It is obvious that most of Canada has inevitably limited ground measurements which tend to not capture the effects of mountainous topography in areas such as the Rocky Mountains which are headwaters for most of the rivers flowing in either side of the Cordillera. As judged by the extensive verification of the simulations you carry out in the paper, it is possible that your study could explore the capability of the current generation of mesoscale atmospheric models such as WRF to reproduce precipitation and temperature features which are important for hydrology and water resources applications in western Canada. Simply, you have to also stress the need for improved forcing products given the data paucity in many regions of Canada.*

**We thank the reviewer's careful examination of our manuscript in terms of technical details and writing structures. We have added a sentence to the objective of the paper in the last paragraph of the introduction to state the objective of evaluation of WRF's capability in western Canada. We also stressed the importance of high quality meteorology data over Canada for providing forcing and evaluation for dynamical downscaling in the abstract, the section for evaluation and conclusion. In the meantime, we revised the abstract to make it more concise. Please see the revised the abstract to see the modification: Lines 35-40**

*Pseudo-Global Warming (PGW): what is the rationale behind employing the PGW downscaling approach instead of the continuous downscaling method often used in region experiments such as NARCAPP and CORDEX? You state in Section 2.2 that 'The second simulation was a climate perturbation or sensitivity experiment following the Pseudo-Global Warming (PGW) approach used in Colorado-Headwaters work (Rasmussen et al., 2014, 2011). Climate projections from GCMs introduce large uncertainties because of the substantial inter-model variability among GCMs (Deser et*

*al., 2012; Mearns et al., 2013), which can obscure the climate change response due to global warming. Using the PGW approach, rather than the intermodel variability, can isolate radiative forcing and its associated circulation as the sole reason for the regional climate response'. What the many disadvantages of the PGW method? See for example: Sato, T., F. Kimura and A. Kitoh, 2006: Projection of global warming onto regional pre- cipitation over Mongolia using a regional climate model, Journal of Hydrology, Volume 333, Issue 1, 30 January 2007, Pages 144-154. Vasubandhu, M. and M. Kanamitsu, 2004: Anomaly Nesting: A Methodology to Down- scale Seasonal Climate Simulations from AGCMs. Journal of Climate, vol. 17, Issue 17, pp.3249-3262*

**We have added several sentences to state the drawbacks of the PGW approach and cited the references suggested by the reviewer. Lines 175-182: "Using PGW methodology during a future period also requires less computation resource than a continuous simulation spanning a century. However, the PGW method also has its disadvantages and limitations. Addition of climate change signal onto the reanalysis field may introduce an imbalance to the lateral boundary forcing because the nonlinear terms are not necessarily additive to balance the dynamics (Misra et al. 2004). PGW also does not fully consider the nonlinear interaction between global warming and atmospheric circulation changes, thus, cannot estimate the changes in future storm frequency, storm intensity, and the positions of storm tracks, which all interact with the large-scale climate system beyond the model boundary and could not represented by simply adding ther- modynamic and kinetic change to current weather and climate (Sato et al. 2007). "**

*Added value of convection permitting downscaling: the overarching aim of this study is to provide high resolution modelling of mesoscale meteorology phenomena over the western Canadian region. However, the authors haven't paid attention to the 'added value' of running WRF at 4km. For example, why should one trust the precipitation simulated by WRF at 4km compared to a low resolution output such as that of the*

[Figure]

*Community Earth System Model (CESM)? This could have been the main stem of this study but it is not addressed here. I urge the authors to address this issue using any other available climate model products or observational products. For example, a validation could be made of WRF-resolved convective precipitation and other available products.*

**The evaluation of added values of CP RCM compared to GCM/RCMs with coarser grid spacing needs observation datasets with high spatial and temporal resolution such as NCEP Stage IV (Nelson et al. 2016) and the Integrated Nowcasting through Comprehensive Analysis (INCA, Haiden et al. 2011) that assimilate observed precipitation from stations, radars, etc. Even for these high resolution datasets, their quality degraded in regions with low station density. It once again reminds us the importance of high density observation network for validation of high-resolution simulation. We have compared the CMIP5 ensemble and ERA-Interim versus observation in addition to WRF simulation. The first columns of Figs. S5, S6 show the temperature and precipitation climatology of CMIP5. The temperature distribution lacks the observed details over the mountainous regions. The precipitation of CMIP5 does not have the dryer interior basin in the Canadian Rockies and the secondary peak precipitation near the front range of the Canadian Rockies.**

*Validation of WRF CTL against observations: I am surprised by the agreement between the WRF CTL simulation and the observed data sets over the domain. The WRF simulations in this study should inherit the biases from its forcing (ERA-interim) nudging was not involved in the downscaling procedure. In such cases, some form of bias correction is often applied to the WRF output. A related question is how good are the ANUSPLIN, NARR (âĹij32km) and CaPA estimates relative to station observations over the study domain? These products are issued at âĹij10 km, so you upscaled 4km WRF to 10 km and downscaled NARR to 10 km for direct comparison? I suppose several other station observations exist especially over the poorly gauged rocky*

*mountain chains which could give us a better insight on the performance of WRF CTL in high elevation areas. Are mountain weather processes (especially in winter) well captured in the ANUSPLIN and CaPA data products? If not, this is a limiting factor to the verification you have shown in the paper.*

**The evaluation of the precipitation datasets over Canada have been conducted by Wong et al. (2017) for various basins. The main conclusion that paper is most reanalysis datasets perform poorly over Canada, including NARR which assimilates precipitation observation. Part of the reason is the lack of observation network. All the data have been interpolated to the WRF 4km grid to do the comparison. ANUSPLIN and CaPA still cannot capture mountain weather processes due to the sparse observation station and the elevation placement of sites. Radar observation is also hindered by the topography. Winter precipitation observation often suffers from under catchment due to surface processes.Limitation of the evaluation is noted in our added text: Lines 220-235.**

*The coarse (âĹij80 km) grid spacing of ERA-interim: The region of Canada where you made the WRF simulations is topographically complex and challenging to model with a coarse DEM and grid spacing of ERA-interim to effectively capture the orographic effects in air temperature and precipitation. Such a coarse resolution could underestimate the complexity of topography in western Canada. Did you check to ensure that the highest peak of Rocky Mountains used in the WRF DEM is comparable to that in ERA-interim? This could also explain a lot of the biases you found in this study.*

**We thank the reviewer for bring to our attention the elevation difference between the surface of each data set. The coarser resolution of gridded observation is the reason there are cold biases over peaks and warm biases over valleys in the Canadian Rockies in WRF-CTL compared to ANUSPLIN, as seen in Fig. 3. ERA-Interim shows a cold bias at the same locations versus ANUSPLIN, which means a finer resolution reduces the cold biases in the ERA-Interim reanalysis in the valleys of southern BC. But the large warm bias in the northwest in winter**

**is not due to the topography but by inheritance of the bias from ERA-Interim. The added discussion is at lines 301-303, 310-312**

*Sensitivity of various schemes and selection methods: the feeling I get after reading through this study and that of Liu et al, 2017 is that most of the schemes used in the model set up over the CONUS were simply repeated for the Canadian domain. This could be probabilistic to model water balance and energy exchanges in more snow dominated high latitudes using the same schemes as over the US. The follow up question is how you went about selecting the combination of parameterization schemes, planetary boundary layer schemes, and radiative transfer schemes for Canada north of 50o? Some results from the diagnostic test runs should be included as supplementary material. This is a study which involved model set ups and numerical experiments, thus the authors should elaborate on their modelling strategy by providing these details.*

*Liu, C., Ikeda, K., Rasmussen, R., Barlage, M., Newman, A. J., Prein, A. F., Chen, F., Chen, L., Clark, M., Dai, A., Dudhia, J., Eidhammer, T., Gochis, D., Gutmann, E., Kurkute, S., Li, Y., Thompson, G. and Yates, D.: Continental-Scale ConvectionPermitting Modeling of the Current and Future Climate of North America, Climate Dynamics, 49(1), 71–95, doi:10.1007/s00382- 016-3327-9, 2017.*

**The choices of schemes are mainly based on previous studies over the Colorado Rockies where snow processes, winter precipitation and mountain precipitations are important. We have added the related references in the description of the model setup. Lines 152-158:**

**"These physics schemes were chosen based past good model performances using these schemes in cold regions (Liu et al. 2011, Rasmussen et al. 2014, Liu et al. 2017). Liu et al. (2011) did a comprehensive sensitivity study on the simulation of winter precipitation in the Colorado headwater region using various physics schemes. They found the Thompson et al. (2008) and Morrison et al. (2009) microphysics schemes have comparable skills and are superior to other**

**schemes. The dependence of performance on land surface, PBL, and radiation parameterizations is moderate or weak due to the weak land surface coupling, shallow PBL, and weak solar radiative heating in the winter (Liu et al. 2011)."**

*Suitability for impact studies: 'For many hydrological and agricultural applications, bias-correction of temperature and precipitation for RCM outputs often need to be reconciled with benchmarked parameters or criteria' Are you recommending some form of bias correction be applied to the data before using it for impact analysis? The narrative throughout the paper is that WRF compared well with observations if not of some wet and cold biases over the mountain areas and along the coast. Based on your analyses, would you recommend that bias correction be applied to the WRF data for the whole domain or only over the regions mentioned above? Figure 3-11 indicate that WRF is quite biased compared to observations if you look closely at the mean error maps. A 4 degree cold bias has serious implications for snow hydrology.*

**We have mentioned the bias in precipitation and temperature in our simulation relative to CaPA and ANUSPLIN. We recommend bias correction to be conducted for applications that is calibrated with observed hydrometeorology. Lines 644-645.**

*Figure 8: taking an ensemble spread of ANUSPLIN, NARR and CaPA to compare WRF against isn't correct. I see this in some studies in the scientific literature but it's ill-informed. These observations are not climate model outputs that respond to the same forcing and can be perturbed to generate an ensemble. The purpose of the verification exercise is also to assess the skill of various observed products and where WRF stands in comparison. Simply computing the spread of the three observational data sets and comparing WRF against doesn't make any sense. Consider comparing WRF against each of these products.*

**We have modified the figure to plot the individual curves of each dataset and rephrase the description in the text referring to the figure accordingly. Please**

**see Fig. 5.**

**2  Minor comments**

/It is not a wise thing to do by not providing continuous line numbers for the manuscript.
I see this in a lot of manuscripts often submitted by newcomers to the field. /

**We now use a continuous line number . The original line number choice is from
HESS's word template.**

- *Abstract: the authors should consider reducing the length of the abstract. What
  do you mean by 'Due to this shift in precipitation intensity to the higher end in the
  PGW simulation, the seemingly moderate increase in the total amount of precipi-
  tation in summer for both the Mackenzie and Saskatchewan river basins may not
  reflect the real change in flooding risk and water availability for agriculture'? This
  kind of ambiguous state- ment isn't suitable for inclusion in the abstract. Con-
  sider revising the abstract in simple terms without leaving the reader having to
  read the body of the manuscript in order to understand the abstract. - Change
  'and for studying climate impacts in hydrology, agri- culture, and ecosystems' to
  'and for studying climate impacts on hydrology, agriculture, and ecosystems' -
  'The change in probability distribution of precipitation intensity also calls for inno-
  vative bias-correction methods to be developed for the application of the dataset
  when bias-correction is required'. This isn't the crux of the paper and shouldn't
  be the closing statement of your abstract. Rather you should highlight the im-
  portance of high resolution modelling compared to other mesoscale downscaling
  approaches which still parameterize convection.*

**We have revised the abstract to be concise and clear. The statement about bias-
correction is moved from the end of the abstract to right after the sentence about**

**the non-stationarity of precipitation. We have rephrased the sentence as the reviewer suggested.**

2.1   Figures

- *The mean error maps have a delta sign beside the colorkey. What does it represent? Some readers may take it to mean a change a precipitation or temperature between two periods.*

**We have added a sentence to clarify the meaning of the delta sign as the difference/bias between WRF-CTL and observation/reanalysis. Please see Figs 3,4,S1-S3.**

- *Write degree Celsius as (oC) in all figures.*

**We have changed the symbol for degree Celsius. Please see Figs. 3,8,S1,S2.**

- *Figure 4: the colorkey to the left is not informative. You need a diverging color ramp with 0 at the centre to represent the 'white' color break. It is not possible to interpret the figure in its current form.*

**We have changed the colour keys and labels. Please see Figs. 3-4.**

- *Figure 6 and 7: the colorbreak labels are not informative. Use equal breaks for both wet and dry*

**We have changed the colour keys and labels. Please see Fig. 4 and S4.**

- *Figure 16: why a log and not a linear scale for daily precipitation?*

**Figure 16. (now Fig. 12) with log for daily precipitation is to see the detail at the high end precipitation.**

Please also note the supplement to this comment:
https://www.hydrol-earth-syst-sci-discuss.net/hess-2019-201/hess-2019-201-AC1-supplement.pdf

**Supplement:**

[revised manuscript text omitted]
 (1st column), WRF-PGW (2nd column), the difference (PGW - CTL, 3rd column), and percentage difference over CTL (4th column). On the right hand side, the projected changes from CMIP5 ensemble for precipitation (2071-2100 – 1976-2005, 5th column) and in percentage (6th column) are shown.**

[Figure]

920

**Figure 10: Annual cycle of temperature and precipitation projected by CMIP5 ensemble. The orange bars indicate the basin average temperature of current climate (1976-2005). The red bars represent the basin average temperature at the end 21st century under RCP8.5(2076-2100). The white bars denote the change in temperature at the end of century relative to the current climate. The dark green bars indicate the basin average precipitation of current climate. The shallow green bars represent the basin average precipitation at the end 21st century under RCP8.5. The white bars denote the change in precipitation.**

925

**WRF west Canada 4-km downscaling**

[Figure]

930 **Figure 11: Annual cycle of temperature and precipitation projected by WRF-PGW (2001-2015). The pink bars indicate the basin average temperature of current climate. The red bars represent the basin average temperature at the end 21st century under RCP8.5. The white bars denote the change in temperature at the end of century relative to the current climate. The dark blue bars indicate the basin average precipitation of current climate. The shallow blue bars represent the basin average precipitation at the end 21st century under RCP8.5. The white bars denote the change in precipitation.**

935

940

[Figure]

**Figure 12: Daily precipitation probability density function in Mackinzie River Basin (top two rows) and Saskatchewan River Basin (bottom two rows) for WRF (CTL, PGW) and CaPA, ANUSPLIN with linear-log, log-log axes for density (y) and amount (x).**

[Figure]

mm/ 3hr

**Figure 13: Three hourly precipitation distribution from station observation in MRB (left) and SRB (right) and those corresponding to WRF-CTL, WRF-PGW simulations.**

[Figure]

**Figure 14: Hourly extreme precipitation frequency density over western Canada from station observation, WRF-CTL and PGW. The bottom panel shows the ratio between PGW and CTL for events with different intensities.**

[Figure]

*Figure 15: Top: Extreme statistics of daily maximum temperature in summer WRF-CTL vs WRF-PGW, 95thpercentile. Bottom:*
*Extreme statistics of daily minimum temperature in winter WRF-CTL vs WRF-PGW, 5th percentile.*

[Figure]

**Figure 16: Extreme statistics of daily precipitation in summer (a) and winter(b) for WRF-CTL vs WRF-PGW, from top to bottom: 90th, 95th, and 99th percentile.**

965

**9 Supplementary**

[Figure]

**Figure S1: The seasonal averaged daily minimum temperature in spring (MAM, 1st row), summer (JJA, 2nd row), and autumn (SON, 3rd row), and winter (DJF, 4th row) from ANUSPLIN (left column) and WRF-CTL (middle column) and the difference (CTL - ANUSPLIN, right column).**

[Figure]

980

**Figure S2: The seasonal averaged daily maximum temperature in spring (MAM, 1ˢᵗ row), summer (JJA, 2ⁿᵈ row), and autumn (SON, 3ʳᵈ row), and winter (DJF, 4ᵗʰ row) from ANUSPLIN (left column) and WRF-CTL (middle column) and the difference (CTL - ANUSPLIN, right column)**

[Figure]

985

**Figure S3: The seasonal averaged daily temperature in spring (MAM, 1st row), summer (JJA, 2nd row), and autumn (SON, 3rd row), and winter (DJF, 4th row) from ANUSPLIN (left column) and ERA-Interim (middle column) and the difference (ERAI - ANUSPLIN, right column)**

[Figure]

990

**Figure S4: The daily mean precipitation in spring (MAM, 1st row), summer (JJA, 2nd row), and autumn (SON, 3rd row), and winter (DJF, 4th row) from ANUSPLIN (1st column) and WRF-CTL(2nd column), and their absolute (3rd column) and relative differences in percentage (4th column).**

995

[Figure]

**Figure S5: The daily mean temperature in spring (MAM, 1st row), summer (JJA, 2nd row), and autumn (SON, 3rd row), and winter (DJF, 4th row) from CMIP5 ensemble 1976-2005 (1st column),  CMIP5 RCP8.5 ensemble 2071-2100 (2nd column) and the difference (CMIP5_RCP8.5- CMIP5_HIS, 3rd column).**

[Figure]

**Figure S6: The daily mean precipitation in spring (MAM, 1ˢᵗ row), summer (JJA, 2ⁿᵈ row), and autumn (SON, 3ʳᵈ row), and winter (DJF, 4ᵗʰ row) from CMIP5 ensemble 1976-2005 (1st column), CMIP5 RCP8.5 ensemble 2071-2100(2nd column) and the difference (CMIP5_RCP8.5- CMIP5_HIS, 3rd column) and percentage difference (4ᵗʰ column).**

---

## Author Comment (AC2) · 13 Sep 2019

**We thank the reviewer for reviewing our manuscript. By addressing the points raised by the reviewer, we have revised the manuscript to make it more concise and focused. We hereby sincerely thank the reviewer for the time and effort conducting a thorough review of the draft.**

[Figure]

**1 Answers to each comment:**

*1. The writing is too wordy and also there are too many figures. Here are a few examples:*

**In general, we have reduced the figure numbers to 15 from 21 by removing and combining figures. We have also revised the abstract to make it more concise.**

/1) The description of the CMIP5-derived perturbation on page 5 is too detailed. I think the authors only need to present the major features of the dynamical and thermodynamical changes./

**We have greatly reduced the description of PGW perturbation from CMIP5. Please see lines 202-215.**

*2) Figures 3-5: I don't see the need for presenting both the daily mean and the daily maximum/minimum temperature because the mean is just an average of the maximum and minimum.*

**We have moved the daily minimum and maximum temperature to the supplementary. Please see the daily minimum and maximum temperature in the supplementary Figs. S1, S2.**

*3) Figures 6-7 can be merged into one. Alternatively, just remove Figure 6 if the authors don't trust ANUSPLIN data.*

**We have put the ANUSPLIN precipitation comparison in the supplementary. Please see Fig. S4.**

*4) Figures 20-21: Because of the great similarity in warming pattern between the three percentiles, there is no need to show all of the corresponding plots.*

**We have opted to show only the 95th percentile for summer and 5th percentile for winter due to the similarities in warming patterns among the percentiles. Please**

**see Fig. 15.**

*2. There are many inaccurate statements, as well as some grammatical errors and typos. Here are just a few examples in the abstract (page 1). 1) L18-19: explicitly resolving cumulus plumes. This is not true. To resolve individual convective elements a sub-kilometer grid spacing is necessary. 2) L19-21: How can you conclude that the simulation agrees with observations in terms of the geographical distribution of cold bias? Logically, this is wrong. 3) L24: "the PGW simulation shows more warming than CTL". The authors may want to say "the PGW simulation shows significant warming relative to CTL".*

**We thank the reviewer to point out these ill-written phrases that slipped through during our editing process. We have revised the sentences mentioned by the reviewer and scrutinized the manuscript to avoid such mistakes. Please see the revision in the abstract.**

*3. Section 4.1. For a fair comparison between WRF downscaling and CMIP5 projection, the temperature and precipitation changes for CMIP5 should be computed as the difference between the 1976-2005 average and the 2071-2100 average, consistent with the climate perturbation used for PGW (i.e., Eq.1).*

**We have calculated and plotted the change from CMIP5 ensemble (2071-2100 - 1976-2005) as the reviewer advised. Please see Fig. 8-9 and Figs. S5-S6.**

*4. I'd like to suggest presenting the temperature and precipitation changes over the whole domain (i.e., Figure13-14) first, followed by describing the sub-domain results (Figs. 11-12).*

**We have made adjustments accordingly and move the paragraphs about the sub-domains behind the whole domain comparison.**

*5. Add CMIP5 projected changes in Figures 13-14.*

**We have added CMIP5 projected changes. Please see the plots in Figs. 8-9 and**

[Figure]

**Figs. S5-6**

*6. Page 5, L8: add "the change of cloud population (Rasmussen et al. 2018)".*

**We have added this reference. Please see line 189.**